# Sensorimotor transformation underlying odor-modulated locomotion in walking *Drosophila*

Liangyu Tao [1], Samuel P. Wechsler[1,2] & Vikas Bhandawat [1] ✉

Most real-world behaviors – such as odor-guided locomotion - are performed with incomplete information. Activity in olfactory receptor neuron (ORN) classes provides information about odor identity but not the location of its source. In this study, we investigate the sensorimotor transformation that relates ORN activation to locomotion changes in *Drosophila* by optogenetically activating different combinations of ORN classes and measuring the resulting changes in locomotion. Three features describe this sensorimotor transformation: First, locomotion depends on both the instantaneous firing frequency ($f$) and its change ($df$); the two together serve as a short-term memory that allows the fly to adapt its motor program to sensory context automatically. Second, the mapping between ($f$, $df$) and locomotor parameters such as speed or curvature is distinct for each pattern of activated ORNs. Finally, the sensorimotor mapping changes with time after odor exposure, allowing information integration over a longer timescale.

Since animals can smell odors emanating from ecologically important resources—such as food and mates that is important for their survival[1]—for long distances from the resource, odors are important for locating and assessing resources. While odors can signal the presence of a resource from a long distance, because of turbulent flow, they provide precise navigational information only in the immediate proximity to odors as odor gradients dissipate a short distance from the odor source[2]. Therefore, finding the source of an odor requires animals to act on incomplete information; an odor encounter will often result in an exploratory search rather than navigational movements[2,3]. During an exploratory search, at each instant, the animal chooses between many possible actions; each action does not result in a defined outcome but might either yield new information or close the door to this information. Many real-world behaviors have a similar structure and require continuous decision-making over extended periods with incomplete information[4]; odor-guided locomotion provides a great opportunity to further understand this important and poorly understood class of behaviors.

Odor-gated exploratory search is characterized by varied and distance-dependent effects on locomotion[5]. At long distances from the source, odors stimulate locomotion and can direct locomotion in an upwind direction; these and other changes in locomotion often bring the animal closer to the odor source[6–12]. At short distances from the source, decreased walking speed, increased turning, and other changes keep the animal close to the resource[13,14]. Overall, animals use a suite of locomotor mechanisms to locate odor sources, home in on them, and interact and utilize resources important for survival[5].

Characterizing odor-driven changes in locomotion is a challenging problem because of two reasons, both related to stimulus control: First, because of the nature of odor transport, the spatial and temporal pattern of odor experienced by the animal is varied and changes with environmental conditions and distance from the odor source; this diversity makes the characterization of the odor stimulus difficult[2,15–18]. Second, how olfactory neurons encode a given odor is dependent not just on odor identity but also on its concentration: Odors are detected by olfactory receptor neurons (ORNs), which each express one to few receptors that determine the ORN's response profile and therefore forms an ORN class. The complement of ORN classes activated by a given odor depends on the odor concentration[19].

[1]School of Biomedical Engineering and Health Sciences, Drexel University, Philadelphia, PA, USA. [2]Department of Neurobiology and Anatomy, Drexel University, Philadelphia, PA, USA. ✉e-mail: vb468@drexel.edu

The first challenge of defining the odor stimulus experienced by the animal was addressed in work done in moths when it became possible to record odor stimulus in flying moths[20,21]. Similar work has been performed in other animals, including other insects[22–24], both in flight[8,25] and in walking[11,12,26–32] revealing many conserved mechanisms at play. These experiments represent significant progress in our understanding. However, two limitations remain. One limitation is relating these mechanisms to a model for locomotion. Attempts to model behavior have suggested that these mechanisms work well when the environment is predictable[33] but may not work well in an unpredictable environment[7]. Another limitation is that few studies have connected behavior to neural response because of the problem of replicating odor stimulus in an electrophysiological rig. These limitations mean that neural algorithms underlying odor-modulated locomotion remain poorly understood.

Another limitation of the work described above is that experiments were performed on a single odor or a single odor blend; therefore, the second challenge of relating the complement of ORNs activated by different odors and behavior remained unaddressed. Relating activation of different ORN classes to the movement of freely locomoting animals has predominantly been addressed only in *Drosophila* because of the genetic tools and the relative ease with which experiments can be performed on many flies. Much of this work has focused on valence or what makes an odor attractive or repulsive[34–37]. To probe the locomotor mechanism that leads to attraction—i.e., how are activities in different ORN classes related to changes in locomotion —we created a ring arena whose center had a fixed odor concentration, and the periphery did not have any odor[13]. Flies only experienced odors when they went inside the central odor-zone; therefore, both the timing and concentration of odor were known. This study showed that different ORN classes activate many different motor parameters independently[13]. Independent control of motor parameters by different ORN classes can be executed by parallel pathways that connect each ORN class to multiple higher-order neurons: ORNs of a given class project to a single glomerulus in the antennal lobe. Each glomerulus is innervated by multiple types of second-order neurons called projection neurons (PNs) including excitatory and inhibitory uniglomerular PNs and multiglomerular PNs, implying multiple parallel pathways downstream of each ORN class[38,39]. Each PN contacts multiple third-order neurons, providing even more opportunities for parallel computations[40].

Although the previous study showed that different ORNs affected motor parameters independently, much of the analysis was based on averages over minutes and did not explain how moment-by-moment firing of ORNs affects a fly's locomotion. Neither did it connect changes in motor parameters to changes in the distribution of the fly in the arena. Equally significantly, locomotion itself would affect sensory experience because the fly's locomotion affects odor sampling: if the fly darts in and out of the odor quickly, the responses to subsequent pulses of odors will be affected by the ongoing response to the first odor encounter. Sampling dynamics play an important role in mammalian odor coding[41].

In this study, we address three unsolved problems. First, we obtain a moment-by-moment record of both the sensory and behavioral information by recording from ORNs and measuring changes in locomotion, respectively. Second, we create a generative model of locomotion to show that the measured changes in motor parameters indeed describe the fly's overall behavior. Finally, we systematically activate multiple patterns of ORN classes to unravel the logic between patterns of active ORN classes and the resulting behavior change. We solve the above problems by optogenetically activating different ORN classes and measuring behavioral changes in *Drosophila*. We also measure the ORN response to the stimulus experienced by the fly. This experimental design provides an accurate estimate of the temporal pattern of ORN activity and the identity of the ORN activated and

allows us to accurately characterize the underlying sensorimotor transformation. We discover that the fly automatically adapts its locomotor strategy on both short and long timescales and that its behavioral response depends on the complement of ORNs activated.

## Results

### Changes in the distribution of the fly depend on the combination of active ORN classes

We focused on subsets of ORN classes activated by a powerful attractant, apple cider vinegar. Apple cider vinegar activates seven ORN classes[13]. We activated five of these ORN classes either singly or in combinations of two or three ORN classes using genetic lines that express the transcription factor *Gal4* under the control of olfactory receptor promoters[42–45]; each genetic line expresses the transcription factor *Gal4* in a single ORN class. *Gal4* was used to drive the expression of CsChrimson (Chrimson for short), a red light-activated channel[46]. Flies that express Chrimson under the control of known ORN classes were placed in a small circular arena (8 cm in diameter) whose central region—a circular region 2 cm in diameter—had a fixed intensity of the light[47] (Fig. 1a). As the fly walked into the region with the red light (light-zone or stimulated region), the red light activated specific sets of ORNs; the resulting behavioral change was assessed. Because flies' photoreceptors have low sensitivity in the long wavelength, their behavioral response to red light itself is small. Chrimson requires retinal to respond to light. Retinal is fed to the flies by raising them on food containing retinal; flies raised on non-retinal food served as controls. As in previous studies[13,47], we first measured the fly's baseline behavior for a 3-min period during which the light in the central area was off. We then turned the light on, and measured its behavior for an additional 3 min.

In all, we activated 13 combinations of ORNs which included single ORN classes activated individually (five different ORNs), 2 ORN classes activated in pairs (3 combinations), and combinations of three ORN classes (two combinations) (Fig. 1). Because *Or42b*-ORNs were activated at the lowest odor concentration[13], the original experimental design was to activate two other ORNs along with *Or42b*—*Ir64a* and *Or92a*—to sample from ORNs that belong to different receptor classes[45]. This experimental design would imply six combinations in all, all of which are in this dataset. We added three more combinations— *Ir75a, Ir64a Ir75a, Ir64a Ir75a Or42b*—to test specific ideas; the ideas that led us to choose these specific combinations were all based on testing particular hypotheses about rules of integration. These hypotheses are irrelevant to the manuscript and will not be discussed further. The conclusions in this manuscript do not depend on the combination of ORNs activated. Apart from these nine combinations, we also activated larger sets of ORN classes by driving Chrimson under the control of *Ir8a*, *Orco*, and *Ir8a* and *Orco* together. These three receptors are co-receptors expressing in a much larger fraction of ORNs than olfactory receptors themselves. In all, we performed recordings from 314 retinal flies and 289,290 control flies to give us 3624 min of data.

Activation of even a single ORN class can change the distribution of flies in the arena. Most combinations that we studied showed some change in the distribution of the flies, such that flies spent more time in and around the central light-zone than at the arena border (Figure S1). Control flies show a small difference in their distribution when the light is turned on. This difference might reflect some attraction to light itself. But the attraction to light is unlikely given that many genotypes do not show any change. The attraction is more likely to be due to a small increase in the activity of the ORNs in the control flies. Regardless, except for *Ir75a, Or42a*, and *Or42a; Or42b; Or92a*, there is a noticeable change in the time spent inside the odor-zone.

However, each ORN combination differentially affected the fly's spatial distribution. Two examples are shown in Fig. 1b: Flies whose *Or42b* and *Or92a* neurons were activated explore the entire light-zone and make frequent forays outside the stimulated region. In contrast,

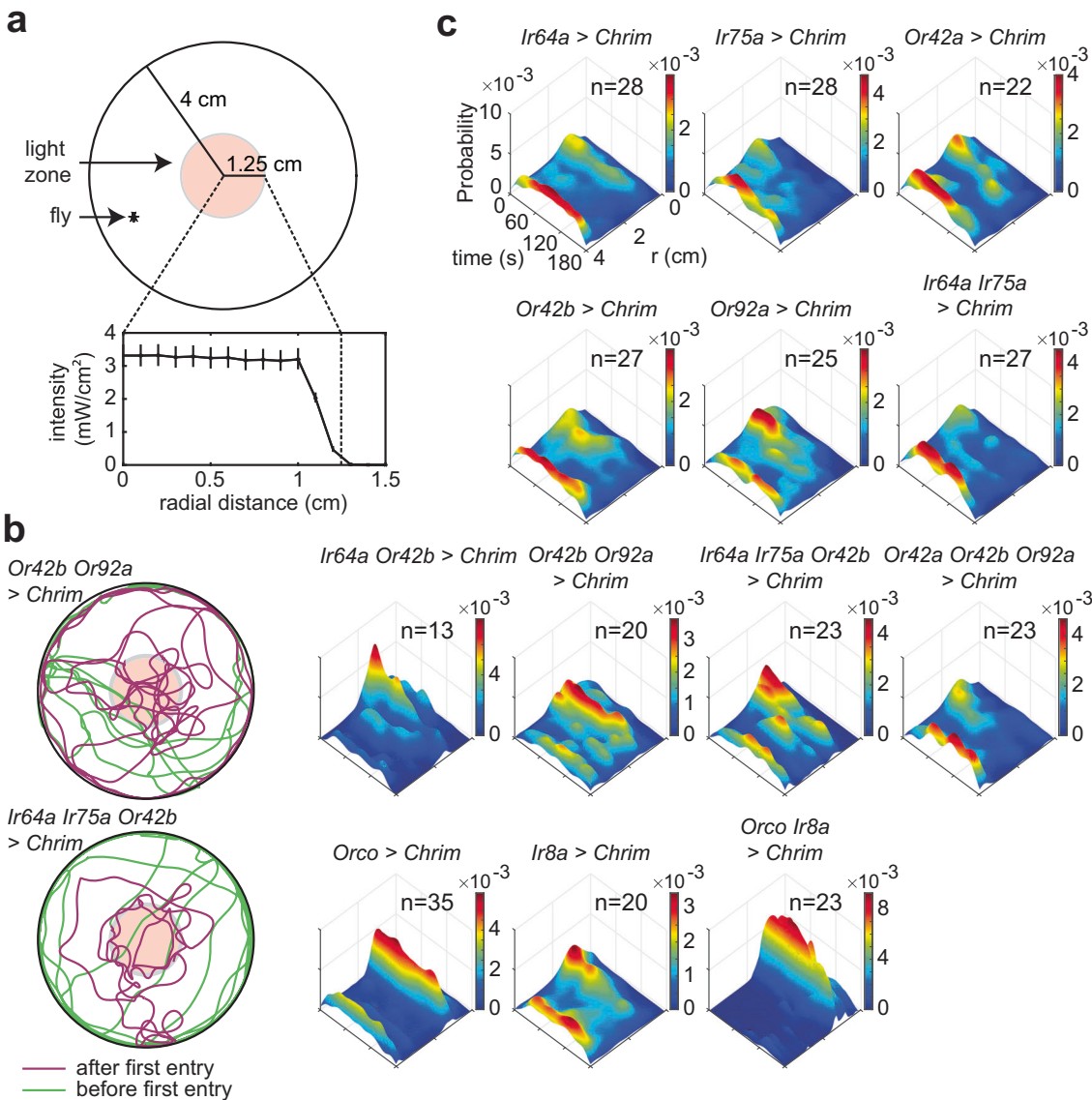

**Fig. 1 | Activation of different ORN classes affect a fly's spatial-temporal distribution in the ring arena in distinct ways. a** Schematic of the arena. Line and error bars represent the mean and range of values measured intensity values. **b** Example trajectories where different ORN combinations are activated show that when Or42b and Or92a ORNs are activated, the flies spend their time more equally throughout the arena. In contrast, activation of Ir64a, Ir75a, and Or42b results in the flies spending more time at the border. **c** Spatio-temporal distribution of flies after first encounter with light. The plots show the probability of finding the fly at a given radial distance from the center as a function of time. Note that the color maps are different for each fly but the Z-axis is kept the same for easy comparison across flies.

flies whose *Ir64a*, *Ir75a*, and *Or42b* are activated stay near the border of the light-zone; the difference in the distribution of these two genotypes is significant (Figure S2). These differences in the fly's behavior can be assessed by plotting how the fly's density in different regions of the arena changes as a function of time after its first encounter with odor (Fig. 1c, Figure S3). Different combinations of activated ORN classes produced different distributions at the light-zone border, within the border, or around the stimulated area. Another difference in behavior is how the fly's behavior adapts over time (Fig. 1c, Figure S4); as an example, behavior downstream of *Ir8a* activation adapts faster such that the flies spend less time inside the stimulus zone at later times in the stimulus (Fig. 1c, also see Figure S4).

Consistent with most previous work[36,48], activating single ORN classes either causes no change in the time spent inside (*Ir75a, Or42a*) or a small change in the time spent inside (*Or42b, Ir64a, Or92a*). Activating two ORN classes produces a larger increase in the time spent inside. The time spent in the vicinity of the stimulated zone when all *Orco* ORNs−consisting of 70% of all ORNs−are activated larger than

the attraction produced by a smaller number of ORNs[36]. Despite the large fraction of ORNs activated by Orco, we observed an even larger change in behavior when we activated both the *Orco* and *Ir8a* ORNs. This stronger attraction is surprising because many studies suggest that activating some ORNs cause attraction and others cause repulsion; based on this idea activating a majority of ORNs would be expected to produce some cancelation between attractive and repulsive ORN classes. Overall, the attraction produced in this arena increases with the number of ORN classes activated; this result is consistent with what others have observed[36,48] and is also consistent with the idea of parallel sensorimotor transformation driving a fly's overall behavior.

## ORN responses are shaped by locomotion, making firing rate a poor measure of sensory experience

Based on the position of the fly's head and the intensity of light at each point in the arena (Fig. 1a), we recreated the intensity of light that a fly experiences as a function of time (Fig. 2a) and replayed the stimulus

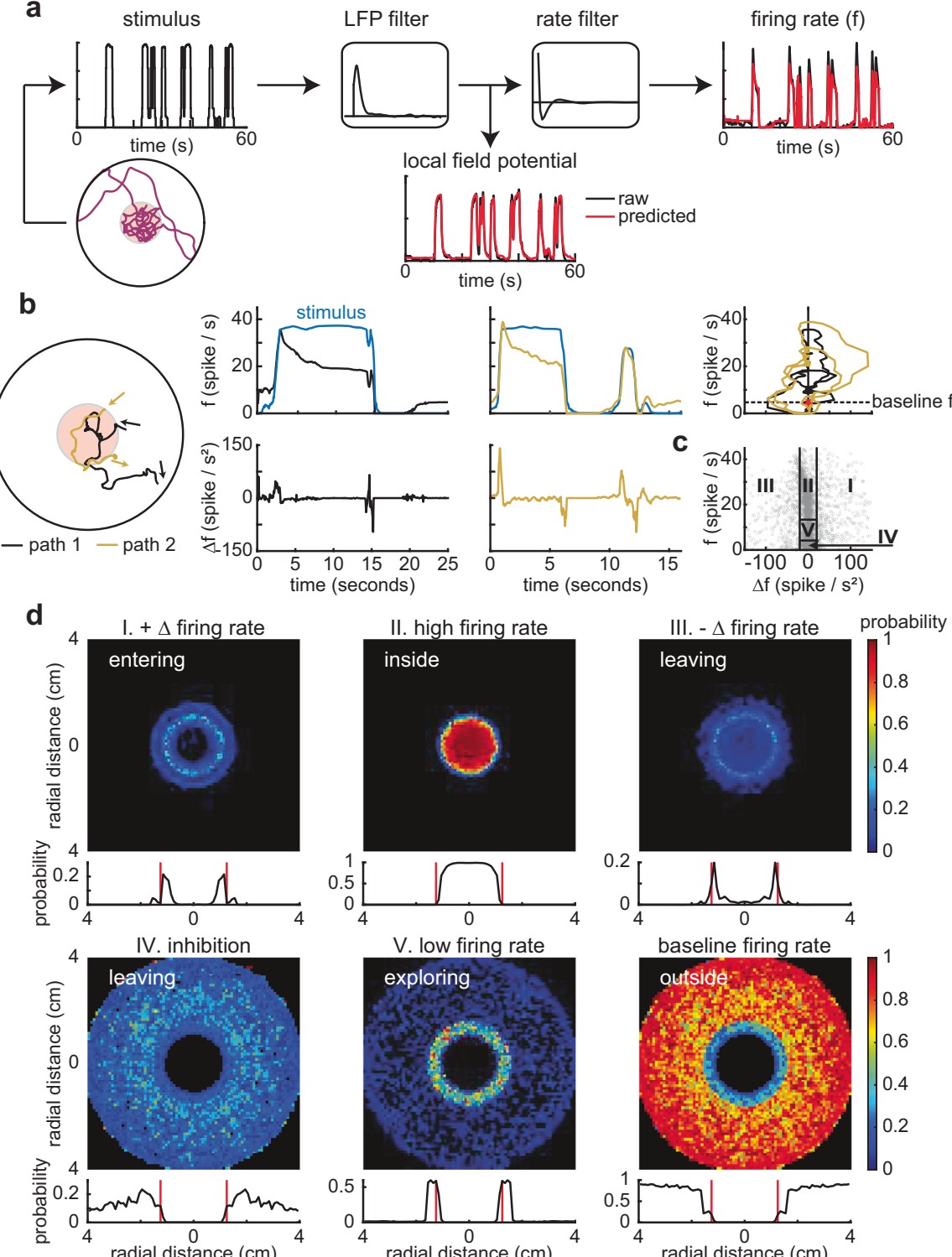

**Fig. 2 | Sensory experience is approximated by immediate ORN firing rate and change in firing rate. a** Linear filters were used to predict ORN responses from behavioral tracks. **b** Two track segments and the firing rate and change in firing rate for the black and yellow show that locomotion affects ORN responses. State space plots showing that sensory experience is a trajectory in the $f$–D$f$ state space (extreme right). Red star indicates the beginning of each trajectory. Dotted line shows baseline firing rate. **c** Randomly selected 10,000 (gray dots) instantaneous firing rate ($f$) and change in firing rate (D$f$). The numbers represent different

regions. Although each entry into the arena results in a unique trajectory though the $f$–D$f$ space, most entries will result in sequential progression from region I to V of the state space. **d** Color maps: The arena was split into 60 × 60 bins where each bin represents the probability of a state-space region (I–V) given the spatial location. The labels−entering, leaving etc. are based on dominant behavior of the fly in each of those states. Probability plots: Probability of being in each region (I to V) as a function of radial distance. The nominal radial light border of 1.25 cm is shown in red.

back to a fly in an electrophysiological rig, to measure ORN responses using single-sensillum recording, a method to perform extracellular recording from the olfactory sensilla[49] that produces a slow current (referred to as local field potential or LFP) likely resulting from the opening of the channels encoded by Chrimson, and the resulting spiking response (Figure S5). We measured responses to 6 different stimulus patterns. Based on these recordings, we created a neural encoder to predict the responses of the ORNs to any arbitrary stimulus profile (Fig. 2a). The neural encoder was a cascade of two linear filters as previously described[50]: the first filter described the relationship between the stimulus and LFP, and the second the relationship between LFP and firing rate (Fig. 2a and S5). There was little difference in the temporal response from different ORN classes (Figure S6). This similarity is consistent with previous work showing that much of the difference in response dynamics results from the odor-receptor interaction, a step that the use of optogenetics bypasses[50]. There is some variability in the amplitude of the ORN responses across ORN classes, but in all cases, the responses are large and are likely to saturate the PNs. To test whether control flies (no retinal) respond to light, as reported by others, we recorded from projection neurons (PNs) downstream to the ORN as they integrate inputs from many ORNs and represent an amplified version of the ORN response. We found that most PNs responded with an increase in activity (Figure S7), implying a small increase in ORN activity in the control flies.

Although the light-zone is static, since the animal enters and exits the light-zone on its own, the time course of stimulus experienced by the fly is complex. The complex stimulus time course, in turn, makes the ORN response profile complex (Fig. 2b): The ORN response when the animal transits quickly through the light-no-light boundary (Fig. 2b, yellow trace) is different from when the fly transits through this boundary slowly (Fig. 2b, black trace). Locomotion also affects the neural response when flies return to the light-zone soon after they exit. In this case, the adaptation from the last excursion to the stimulated region affects the current response, and the peak response is lower. Therefore, the fly's sensory experience is dynamic, and its behavior is likely modulated by recent stimulus history. To model this sensory motor transformation, we first started by assessing whether we can describe the transformation between ORN firing rate and kinematic parameters such as speed or curvature. This approach did not work for most flies (Figure S8) as either the temporal structure of the filter did not make sense or the predictions based on the filter were poor. Next, we tried approaches successfully employed to model the behavior of *Drosophila* larvae: We first used reverse-correlation to estimate the relationship between ORN responses and sharp turns[51,52]. The analysis results made qualitative sense and was consistent with previous work[53] as the turn probability increases when the spike rate or the stimulus decreases (Figure S9). However, because the spike distribution was non-Gaussian, this approach cannot be employed to analyze the data collected in this study quantitatively as the resulting filter would be erroneous, a well-known limitation of reverse-correlation approaches[54]. Next, we tried another approach that was successful in describing behavior in larvae[55] where a logistic Generalized Linear Model (GLM) was employed to predict the relationship between ORN responses and behavior. This approach failed to describe the data (Figure S10). This failure is likely due to the fact that this GLM is instantaneous and has no history. It is possible that GLMs that incorporate time history might describe the data better. Finally, we focused on obtaining linear filters that seek to model the average kinematics during a state based on the firing rate before the state transition. These filters failed to predict the time averaged state trajectory kinematic parameters (Figure S11). Thus, commonly employed methods failed either because the assumptions made when employing those data did not hold for our data, variability amongst flies, and the fly's behavioral response to the same stimulus.

Because the previously employed methods failed, we decided to take a novel approach. Analyses aimed at estimating the linear filter gave us a hint. Because the filters that we obtained as the fly is leaving or entering were transient and returned to 0 within 200 ms, and we described the fly's behavior as it entered or exited the arena quite well. But, these filters failed to explain the fly's overall behavior. We reasoned that the fly's behavior is driven by recent ORN activity, but the relationship between ORN activity and behavior changes with the time it spends inside the stimulated region. This dependence can be modeled by making the animal's behavior dependent on the immediate ($f$) and change in firing rate ($\Delta f$) history. Each entry into the arena is characterized as a trajectory in the ($f$, $\Delta f$) state space (Fig. 2b). Different entries into the light-zone often led to different neural responses (Fig. 2B2), which can be described as different trajectories through the ($f$, $\Delta f$) space. Each entry into the arena goes through a similar transition, starting with a high firing rate and rapid increase in firing rate (region I, entering) to a high firing rate with adaptation (region II, inside) to a high firing rate with a decrease in firing rate (region III, leaving) to inhibition (region IV, left recently), and finally low firing rate (V, exploring border). Therefore, ($f$, $\Delta f$) contains an approximate representation of the sensory experience, and it is a computationally inexpensive method for keeping track of odor history.

The usefulness of ($f$, $\Delta f$) is further illustrated in the spatial distribution of the flies in each of these regimes (Fig. 2d), i.e., where are the flies when they are in a particular region of the $f$–$\Delta f$ state space? Suppose the fly was using just the firing rate at the same radial distance near the odor border, the fly could have a very different firing rate depending on whether the fly is entering in or exploring the boundary. However, taking both $f$ and $\Delta f$—which together describe the different regimes (I–V)—allows the flies to parse whether it is entering the stimulus region, within it, exiting the arena, or was in the arena recently.

In sum, the flies in this arena enter the stimulated zone multiple times. The ($f$, $\Delta f$) is a time history of its sensory experience that starts with its most recent entry to a few seconds after its exit when the ORN firing rate returns to baseline. Therefore, we modeled the behavior as a transformation from recent ($f$, $\Delta f$) to the behavioral parameters. In the next section, we will model how the mean and variance of kinematic distributions change with ($f$, $\Delta f$). This approach has the advantage that it will accurately model these kinematic changes.

## Sensorimotor transformation is probabilistic, dynamic, and depends on the population of ORNs activated

We assume that ($f$, $\Delta f$) in a short time window affects the fly's behavior which we model using an agent-based locomotor model[47]: The agent can be in one of four states (Fig. 3a). Each state is defined by 2–3 parameters that remain constant during the state (Fig. 3b). Note that only the major transitions are marked in Fig. 3b; other transitions occur infrequently; no transitions are disallowed. Therefore, the locomotor model is a probabilistic model with ten parameters in all (enumerated in Fig. 3b)—two of these are at the boundary, so they are not directly affected by ORN activation, leaving eight parameters that are affected by ORN activation (Fig. 3b). The effect of ORN activity is modeled as a change in the distribution of these parameters (Fig. 3c). Specifically, $f$ and $\Delta f$ in a short time window before the fly transitions to a new state determines the parameter distribution in the subsequent state (Fig. 3c). A mapping from ($f$, $\Delta f$) to the probability distribution of each of the eight locomotor parameters describes the sensorimotor transformation. We estimate this mapping for each location in the ($f$, $\Delta f$) space (the regions I to V in Fig. 2 are used only to describe the data and the rules of integration between ORN classes).

To estimate how $f$ and $\Delta f$ affect the distribution of the eight parameters, we binned all instances of the start of a state when the $f$ and $\Delta f$ in the preceding 200 ms were similar; the resulting distributions were well fit by lognormal distributions (Figure S12). A lognormal

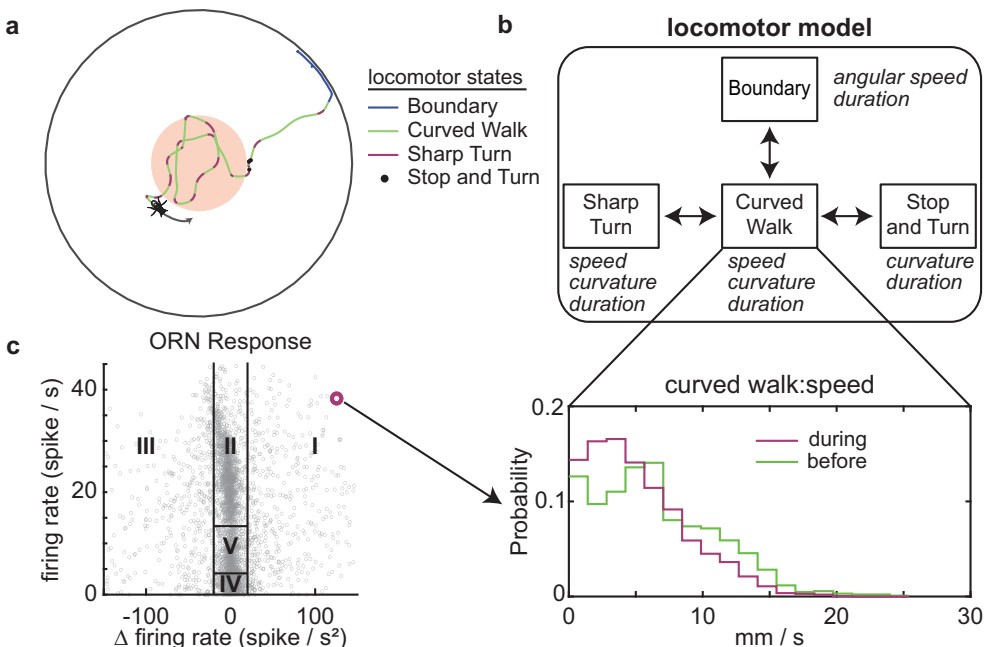

**Fig. 3 | Framework for sensorimotor transformation−(*f*, Δ*f*) in a short time window before state transition affects the distribution of locomotor parameter. a** A fly's locomotion can be described by a 4-state model consisting of sharp turns, curved walks, stops, and the arena boundary (adapted from Tao et al., Plos Computational Biology 2020). **b** Each state is described by just 2−3 parameters, which are listed (in italics) adjacent to the state. As an example, the parameters for curved walk are speed, curvature and duration. The agent selects the parameters for a given state from a distribution; the distribution for curved walk speed is shown (bottom panel). ORN stimulation affects locomotion by altering the distribution. There are 8 such distributions inside the light-zone as well as 2 boundary distributions. **c** The effect of ORNs is modeled as a mapping (represented by the arrow from the red point) between (*f*, Δ*f*) and each of these eight parameters. We will estimate the distributions (for each parameter at each location in the (*f*, Δ*f*) space.

distribution is defined by its mean and variance; using the K-nearest neighbor (KNN) framework, we estimated the mean and variance of this distribution (Figure S13 and "Methods"). In describing the effect of ORN firing on locomotion in Fig. 4, we will focus on the mean of the distribution; the entire distribution is employed in generating synthetic flies (see below). Despite the large number of flies we investigated, we are still data limited in some instances. This limitation means that although we can capture the essence of the sensorimotor transformation between ORN activation and behavior, but likely miss some finer details.

An example of the mapping between (*f*, Δ*f*) and locomotor parameters is shown in Fig. 4a. The speed of the fly during curved walks decreases when the fly is walking along the border of the light-zone (Region V). The same effect on speed is observed during sharp turns (Fig. 4a, regions of significant change are shown in Figure S14). In contrast to the change in speed, the largest change in sharp turn curvature occurs when the ORN firing rate is decreasing (Region III, as the fly is exiting). The change in curved walk curvature is largest in Region V when the firing rate is steady and decreases when there are changes in the firing rate. Finally, the effect on the duration of sharp turns is negligible, while the curved walk durations decrease as the firing rate decreases. These different effects of (*f*, Δ*f*) on locomotor parameters reflect a change in a fly's motor program as it spends more time inside the stimulated region.

The effect of different combinations of ORN classes on a given motor parameter is also different (Fig. 4b). As an example, consider the effect of different combinations of ORN classes on sharp-turn curvature. Activation of single ORN classes *Or42b, Or92a, Ir64a,* or *Ir75a* has a small effect on the curvature of the sharp turn. Activation of both *Or42b* and *Or92a* ORN classes together results in a large increase in the curvature of the sharp turn suggesting a strong additive effect. This additive effect is not observed in the activation of *Ir64a* and *Ir75a* together (Fig. 4b). When three ORN classes−*Ir64a, Ir75a,* and *Or42b* are

activated together, there is a large effect on the curvature of the sharp turn, suggesting that the rules of addition are non-linear and depend on the co-activated ORN classes. The different rules of addition are further highlighted by the effect of different ORN classes on the speed of curved walk (Figure S15). In contrast to sharp-turn curvature, activating a single ORN class−*Or42b*−has a large effect on the speed during curved walks. Activating two ORN classes has an even larger effect on speed. The effect on speed is less obvious when more ORN classes are activated (Figure S15). These data are consistent with our previous work and show that the addition of more ORN classes does not simply scale the observed changes in parameters[13]. We will revisit the rule of integration more quantitatively later in the study (see below).

As mentioned, the ORN responses following entry into the stimulus zone to a few seconds after its exit is represented as a trajectory in (*f*, Δ*f*) space. A new trajectory starting at the baseline begins every time the fly enters the stimulus zone. Thus far, we have investigated the effect of ORN firing that averages the change in behavior across multiple entries. Although the ORN firing rate returns to baseline every time the fly exits the stimulus zone, there may be changes in the sensorimotor transformation across different entries into the stimulus zone. To assess these dynamics, we used the K-nearest neighbors (KNN) approach to assess how the sensorimotor transformation changes with time by evaluating the relationship between (*f*, Δ*f*) and locomotor parameters at different times after the first odor encounter (Figure S13 for methods). We found that the changes in locomotor parameters with time were distinct for different motor parameters; the effect of ORN activation on a given parameter can increase with time, decrease, or stay the same (Figure S16). An example where the effect of ORN activation builds up over time is the speed of curved walks. The prominent decrease in walking speed in the arena center when both *Or42b* and *Or92a* ORNs are activated becomes more pronounced with time but stabilizes (Figure S16A). In the same flies−when *Or42b* and

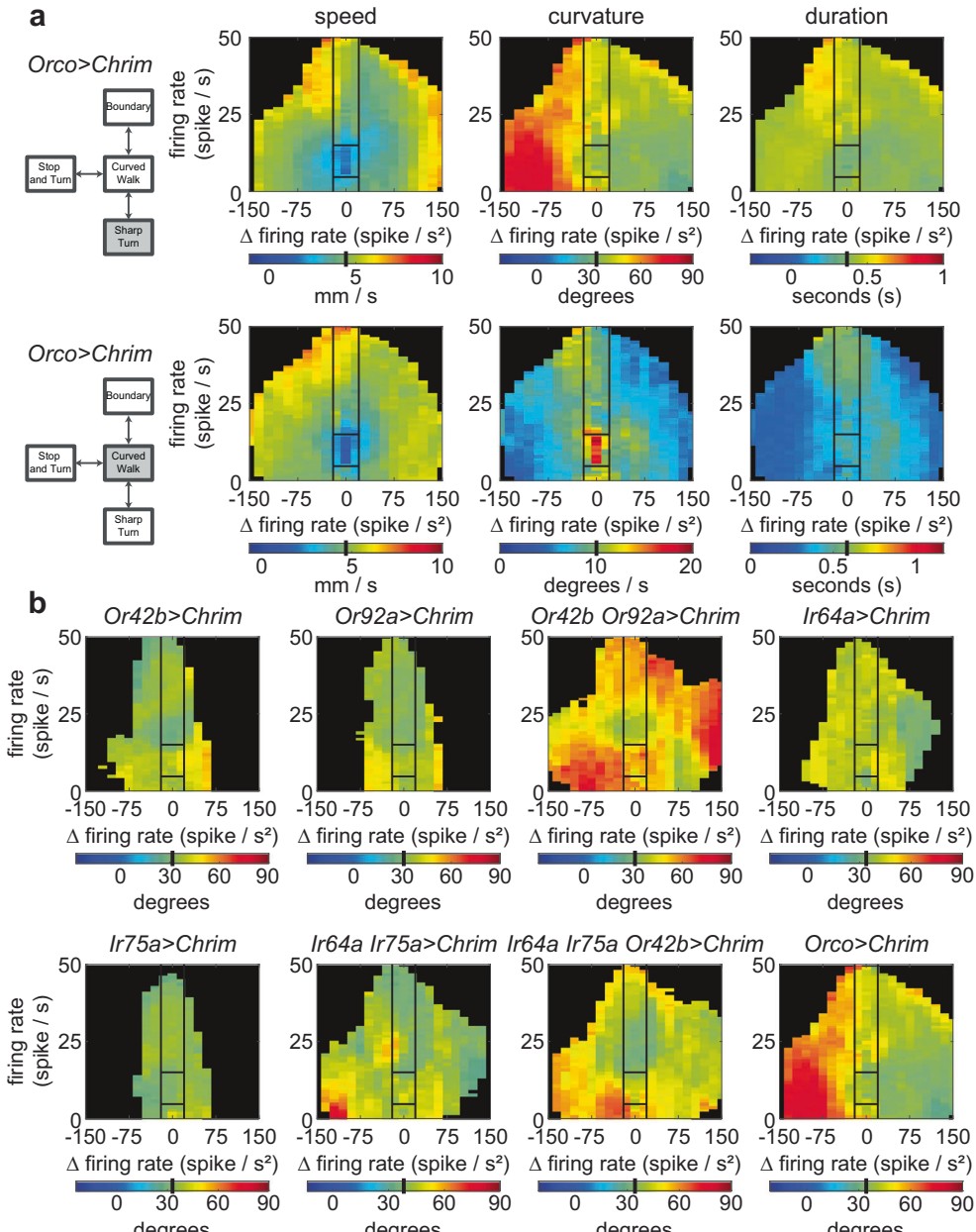

**Fig. 4 | Sensorimotor mapping for each locomotor parameter depends on both the parameter and ORN combination activated.** The mapping between ($f$, D$f$) and distribution of variables was estimated. The effect on mean is plotted here. In all panels, black lines separate out the sensorimotor mapping into 5 broad regions based on firing rate and change in firing rate as described earlier. Baseline means for a given parameter before odor on is shown as a black bar in the colormap. **a** Effect of a given combination of ORN class on locomotor parameters; sharp turn and curved walk speed, curvature, and duration. Different parameters are affected differently. **b** Effect of different ORN classes on a given motor parameter—sharp turn curvature—is different.

*Or92a* ORNs are activated—there is a rapid decrease in the effect of ORN activity on sharp turn curvature, which stabilizes over the duration of the stimulus (Figure S16B). An example where the effect of ORN activation decreases with time is the effect of *Ir8a* activation on the curvature of sharp turn; the effect of activation is largest right after the fly experiences ORN stimulation for the first time and decreases thereafter (Figure S16C). Overall, the sensorimotor transformation for different parameters evolves differently over time.

In previous studies using both odors and optogenetics, we had shown that odors or ORN activation influenced behavior not only when the stimulus is present but also outside the stimulus-zone[13,47]. Some of this effect is likely due to the strong effect of the ORN-off-response (Region IV). Is the effect of ORN activation persistent after the ORN firing rate has returned to baseline? Indeed, when we analyzed

kinematics only during the period when the firing rate was at baseline levels, we found that for some genotypes, there was no change in kinematics during the baseline period (Figure S17). For other genotypes, there are changes in kinematics even when the ORN firing rate has returned to baseline (Figure S17). These changes also adapt over time.

## Flies can turn preferentially at the border to either stay inside or turn inwards

Apart from kinematics, insects can use different forms of directional information to direct their turns toward an odor source[56–59]. We have already shown that flies use directional information in the ring arena[47]. This use of directional information is also evident in the tracks of flies (Fig. 5a), which show flies weaving in and out of the light zone. Flies of

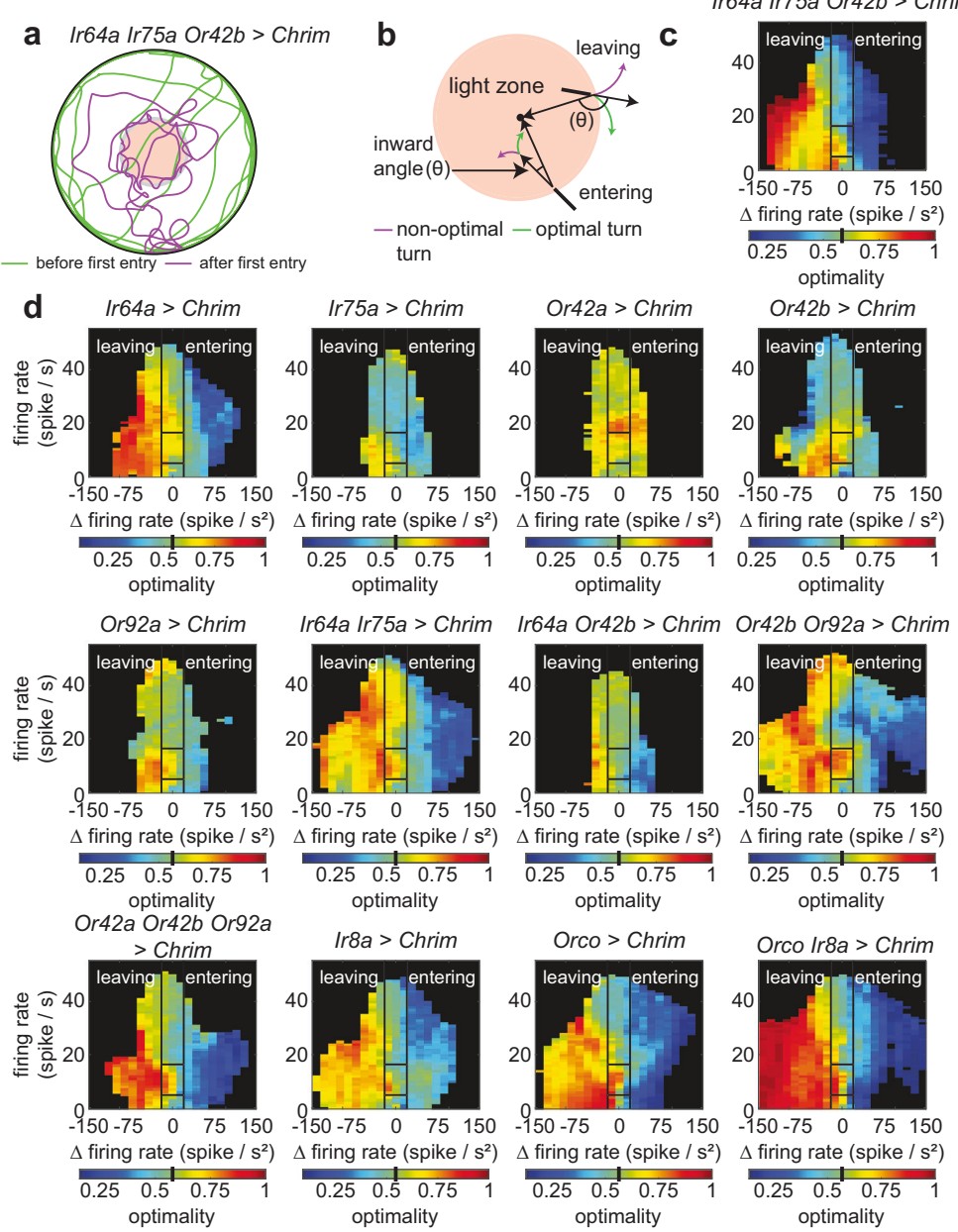

**Fig. 5 | ORN activation alone allows flies to perform directed turns. a** A sample trajectory shows that flies are able to track the stimulus border. **b** Schematic showing optimal and non-optimal turns. Turn direction that decreases the angle between the current direction of the fly and radial vector (or q) are optimal. **c** Turn optimality (defined as the percentage of optimal turns) for Ir64a Ir75a Or42b > Chrimson relative to baseline. Black lines separate out the sensorimotor mapping into 5 broad regions based on firing rate and change in firing rate. Baseline turn optimality is shown as a black bar in the colorbar. **d** Turn optimality for all other combinations of active ORN classes (see text for details).

some genotype—*Ir64a*, *Ir75a*, and *Or42b* are activated together, or where *Orco ORNs* are activated, or *Orco* and *Ir8a ORNs* are activated appear to show a greater propensity for this weaving behavior. We quantify how flies use directional information by quantifying turn optimality (see "Methods"). As the fly exits the light-zone, unless it is moving exactly in a direction normal to the light-zone, it can either turn in a direction where a small turn will bring it inwards (optimal direction) or in a direction in which a large turn is necessary (Fig. 5b).

Similarly, as the flies enter inside, the flies can turn in an optimal or non-optimal direction (Fig. 5b). Indeed, flies prefer turns in an optimal direction (Fig. 5c). They turn towards the border as they exit the arena with a higher-than-chance probability. In contrast, the flies turn back out of the light zone when they are inside. The combination of high

optimality when leaving and low optimality when entering results in the observed weaving behavior near the light border.

This directional preference also depends on the genotype. When a single ORN class is activated, the flies do not exhibit this directional preference except when Ir64a-ORNs are activated (Fig. 5d). Consistent with complex rules of integration observed for kinematic parameters, when both Ir64a-ORNs and Or42b-ORNs are activated, the directional preference becomes smaller. All other combinations in which two ORN classes are activated show an increased propensity for directed turns. Finally, consistent with the border-hugging tracks, when *Ir64a*, *Ir75b*, and *Or92a* ORNs are activated together, turn optimality is particularly large (Fig. 5c). Flies not fed retinal have a smaller turn optimality (Figure S18). However, *Orco*, *Ir8a* displayed similar border-hugging

behaviors when not fed retinal, which is reflected in their turn optimality; this behavior likely reflects weak ORN activation.

The most likely mechanism for directional preference is comparing response between the two antennas[58,59]. We re-analyzed a dataset from Orco-ORNs-activated flies whose right antenna was removed to test this idea. These flies continued to display the ability to perform tight border weaving patterns (Figure S19A). Indeed, these flies could still perform the same type of optimal turns when leaving and non-optimal turns when entering, which suggests that directional preference cannot be explained purely through bilateral comparison of sensory experience (Figure S19B). Flies are likely using temporal comparisons as a directional cue. Given that temporal comparisons are likely driving the increased sharp turn at the odor border, it is also possible that temporal comparisons also drive optimal turning.

We also evaluated whether the flies perform optimal turns through idiothetic path integration by evaluating optimal turns after the ORN activity has returned to baseline level[56]. We found that, indeed, when *Orco* ORNs are activated, there is a higher-than-chance turn optimality irrespective of radial distance away from the center of the arena (Figure S19D).

## A model for diverse rules of integration between ORN activation and change in locomotor parameters

The data presented in Figs. 4 and 5 points to three features underlying sensorimotor integration in the olfactory system: (1) ORN activation affects multiple motor parameters, (2) the effect depends on both $f$ and $\Delta f$ and on the identity of the active ORNs, and (3) is distinct for different motor parameters. The neural substrate for this transformation exists in the fly's olfactory system. We postulate that the different kinds of PNs−excitatory and inhibitory uniglomerular PNs and excitatory and inhibitory multi-glomeluluar PNs−represent parallel channels of communication. Because the microcircuitry underlying each PN type is distinct, they likely have different dynamics in response to the same ORN input[40]. Higher-order neurons, such as lateral horn neurons, can integrate across different combinations of PNs to affect changes in different motor parameters. These ideas are illustrated in Figure S20. We develop a conceptual framework for this parallel sensorimotor integration by investigating the rules of integration in the five regions of the $f$−$\Delta f$ space (Figure S20). Each entry into the stimulus zone will lead to the sequential activation of these output channels. Each channel can have a different effect on a given motor parameter that can be modeled as a change in the distribution (Figure S20). Signals from different ORN classes are integrated according to different rules in each of these five output regions. The integration rules were modeled using a regression model (Figure S21 and "Methods").

We applied this approach to all combinations of ORN classes and illustrated this analysis with how activities from *Or42b-ORNs* and *Or92a-ORNs* affect sharp turn curvature (Fig. 6a). Activation of *Or42b-ORNs* has a large effect (>40% increase in Region I) as the fly enters the arena. This effect is transient, as reflected by the small effect in Region II when the fly is fully inside. The effect on curvature returns when the fly is exploring the odor border (Region V), and when the ORN response is inhibited (IV). Activation of *Or92a-ORN* alone also has a similar effect. When the two ORNs are activated together, there is a large synergistic effect on the curvature of sharp turns except in region I. The effect of this synergism is that the sharp turn curvature is high in all regions when both ORNs are activated. The synergy between the two ORNs means that the effect due to the two ORN classes together is 20–40% higher than expected from a linear sum (Fig. 6a, rightmost panel). Interestingly, although there is still a large effect of the combined ORN activation on sharp turns when the fly is entering the arena (Region I), the interaction effect is not synergistic as the observed effect of the two ORN classes is about 15% smaller than expected from a linear summation.

How activities from two ORN classes are summed depends on the motor parameter: Consider the effect of *Or42b* and *Or92a* ORNs on curved walk speed. In some regions (I, III, IV), the effect of the two ORNs adds sub-linearly such that the increase in speed is not as large as expected from a linear summation. The large decrease in speed observed in region II, when individual ORN classes are activated, is completely abolished when both ORN is activated; this is an example of antagonistic reduction (Fig. 6b).

Interaction terms show other summation rules: one ORN can have a dominant effect, particularly when the two ORN classes individually have different effects; the combined effect of two ORNs can affect individual parameters when neither has an effect individually (not in this case, see Fig. 6c for an example). This diversity of rules is evident in the effect of *Or42b-ORNs* when combined with different ORN classes (Fig. 6c). For a given parameter, the effect depends on the region of the $f$−$\Delta f$ space. For example, take either the curved walk or sharp turn speed: In region II, activating Or42b in combination with any of the other ORNs increases the speed. This increase is observed even though *Or42b* activation alone reduces the speed in region II. In contrast, in regions I and III, *Or42b* activation results in a less pronounced increase in speed. Finally, during the inhibition epoch (region IV), the effect of *Or42b* appears muted, and the overall behavior is close to the behavior due to *Ir64a-ORNs*.

The same diversity applies to the effect on curvature during walks. The result is different both for different regions of $f$−$\Delta f$ space and sharp turn and curved walk curvature: In regions I and III, i.e., when the flies are leaving or entering, *Or42b* activation has an antagonistic effect on the curved walk curvature such that the summed result is smaller than what would be expected from a linear sum (Fig. 6c, bottom right panel, hashed blue). In some cases, the net effect of activating Or42b is so strong that the change in curvature due to Or92a or Ir64a activated alone is almost abolished (green in Fig. 6c, bottom right). In contrast, the sharp turn curvature increases in many regions when Or42b is activated along with the other combinations.

This diversity of integration rules and its dependence on both ($f$, $\Delta f$) makes sense in light of the fact that the motor program should change with the fly's sensory experience. This diversity can not only be supported by olfactory processing circuits but is the only possibility given the widespread convergence and divergence of olfactory signals in higher olfactory circuits. To illustrate this idea, we follow only the most salient feedforward connections from the *Or42b-ORNs*−connections from *Or42b-ORNs* to the uniglomerular PN that directly connects to *Or42b-ORNs* (DM1uPN) to lateral horn neurons−using the recent connectomics data[40] and find that it signals to at least five different LHONs, which all integrate input from different ORN classes (Figure S22). These LHONs can, either individually or in different combinations affect different motor parameters. This connectomic analysis is very limited. Overall, the simple connectomic analysis illustrates how just the connections to one brain region−lateral horn- can subserve the diverse integration rules we discovered here. Given that there are multiglomerular PNs that also integrate inputs from these ORN classes and signal to the lateral horn, as well as other brain regions, such as the mushroom body that receive inputs from antennal lobe and are connected to each other with dense recurrent connections, circuit architecture that might underpin these rules of connection, are very much present in the fly brain. It is important to note that this connectomic analysis is not meant to assert an exclusive role for lateral horn in the sensorimotor transformation outlined in this study.

## A generative model for the effect of odors on locomotion

The analysis above (Figs. 4−6) shows that the effect of ORN activation on locomotion depends on the identity of the ORNs activated. Can the changes in kinematics explain the changes in the distribution of the flies? To this end, we created synthetic flies based on our agent-based model[47]; the details are in the "Methods" section and in Figure S23.

**Fig. 6 | Rules of integration between ORN classes are diverse and depend on region of the state-space and the locomotor parameter. a** Rules of integration for the effect of Or42b, Or92a on sharp turn total curvature. The three panel on the left are the effect on sharp turn. 1 = no effect; 1.46 is a 46% increase. The rightmost panel shows the interaction. Color implies that the interaction was synergistic such that the observed effect is at least 10% larger than for linear summation (orange), and at least 10% smaller (blue). **b** Same as (**a**), but for curved walk speed. The same

two ORN class can have either synergistic or antagonistic interaction depending on the locomotor parameter. Synergistic and antagonistic interactions can be further delineated by whether the individual ORN activation results in an increase (solid) or decrease (hashed) in locomotor parameter. **c** Interactions between Or42b and three other ORN combination Or92a, Ir64a, and Ir64a;Ir75a show that rules of summation between the same ORN classes are diverse. Numbers represent interaction gain effects on locomotion (see "Methods").

Briefly, just like the experimental flies, each synthetic fly walked for 6 min−3 min before the light turned on and 3 min following the light on. Synthetic flies started in the curved walk state at the center of the arena and moved around the arena through a series of transitions into the four states. Curved walks end in a stop, sharp turn, or at the boundary. Tracks corresponding to each transition were generated by sampling from speed, curvature, and duration distributions for each state for the $f$ and $\Delta f$ during the 200 ms preceding each state transition. Using the position of the synthetic flies as a function of time, we estimate the intensity of light experienced by the fly as a function of time; this light intensity was converted into ORN spike rate using the

two-stage linear encoder derived in Fig. 2. The resulting $f$ and $\Delta f$ were used to determine the kinematic distributions that the synthetic flies sampled from at any given time. The duration that each transition lasted was also selected from the empirical distribution.

To assess how well the behavior of the synthetic flies replicated that of the empirical flies, we first compared the flies whose *Orco-ORNs* are activated, a genotype that we have analyzed previously[47], and because these flies show a large change in their distribution. The radial distribution of the empirical and synthetic flies before and after the central red light is turned on is shown in Fig. 7a. As expected, the radial distribution of the synthetic and empirical flies before the light is

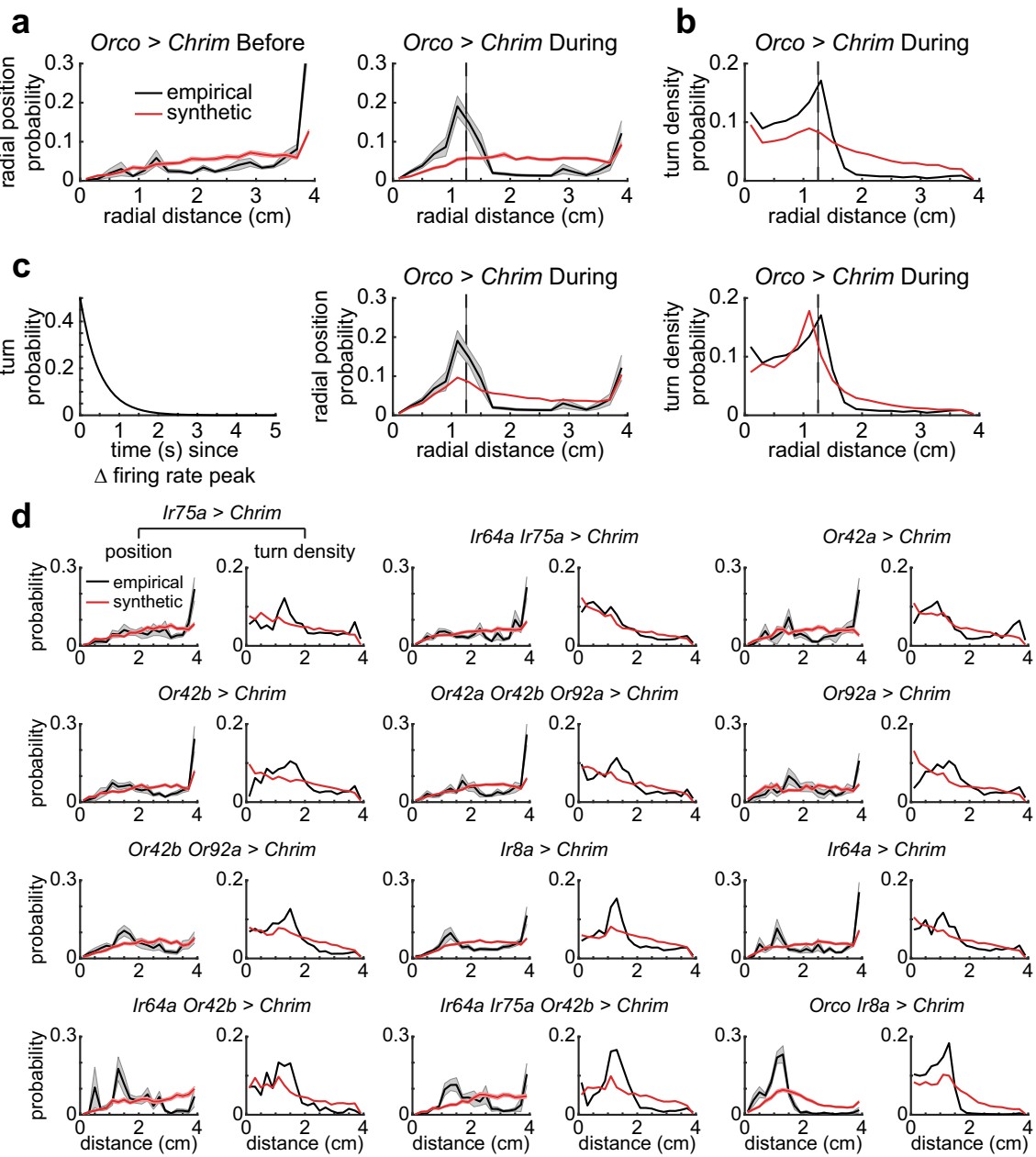

**Fig. 7 | An agent-based model of flies based on locomotor kinematics and turn optimality can capture much, but not all of the flies behavior.** The reasons for discrepancy are elucidated in Figs. S23–S27. **a** Spatial distribution of the empirical and synthetic flies are similar before the stimulus is turned on. During stimulation, the synthetic flies spend less time than the empirical flies inside the stimulated area but more time between the stimulated zone (dashed line) and the outer boundary. **b** The difference in radial distribution results from a larger propensity to turn at the border of the stimulated zone than the synthetic flies. **c** An exponentially decaying

probability of transitioning into a turn was used to model the empirical turn density (right). By improving the turn density fit, the model was capable of better capturing the spatial position (middle) of flies in the arena. **d** When there is no or small increase in turn density at the border, the synthetic flies have a distribution close to the empirical flies. Examples are Ir64a alone, and Ir64a and Ir75a activated together. As the turn density increases, the pattern observed in the Orco activated flies is observed. Radial occupancy in (**a**), (**c**), (**d**) show mean +/− SEM.

turned on is similar. The distributions of the empirical and synthetic flies during the stimulation period are clearly different (Fig. 7a). This mismatch is not due to large differences in kinematics, as many of the kinematic changes in the empirical flies are replicated in the synthetic flies (Figure S24). A closer examination of the radial density in Fig. 7a shows that the crucial difference is that the synthetic flies spend more time in the region between the outer border and the light zone than the empirical flies and lack the large peak in radial density at the border of the stimulated region. These differences imply that we do not fully capture the flies' behavior at the light-zone's border. This result is

consistent with our previous work, where an assumption that the fly changed state each time it crossed the light-zone was necessary to fit the data[47]. Indeed, we found that a negative peak in Δf caused a large increase in the propensity of the flies to exit curved walks and enter sharp turns (Figure S25). Although (f, Δf) capture this tendency—the curved walk duration is lower when Δf is negative (Fig. 4a)—this tendency is underestimated because of estimation errors in the KNN model originating in the lack of data for high negative Δf. Consistent with this idea, there is a large peak in the propensity to turn in the empirical flies that are not captured in the synthetic flies (Fig. 7b). We

attempted to recreate this increased turn propensity by adding a "border choice" parameter. Although we were not wholly successful in matching the turn density observed in empirical flies, we show that incorporating the increased turn density in the model for synthetic flies results in a large peak in the radial density near the light-zone and decreases the density in the unstimulated, non-border regions (Fig. 7c). Another method to increase the turn density at the border was to employ linear filter predictions at the boundary, and the KNN predictions everywhere else. Indeed, using this method both the turn density and radial density of the synthetic flies match that of the empirical flies (Figure S26).

The trend observed in *Orco-ORNs*-activated flies is observed in other genotypes. In flies in which there is a small peak in turn density shows a close match between radial density in the empirical and synthetic flies (Fig. 7d). Combinations of ORNs that increase turn density show the same trend as seen in the *Orco* flies—peak radial density at the border of light-zone that is not captured in the synthetic flies and increased density in the synthetic flies just outside the light-zone. As an example, for *Ir64a; Ir75a*, the discrepancy in turn density is small, and so is the difference between empirical and synthetic flies. Conversely, for other genotypes such as *Ir64a;Ir75a;Or42b*, the discrepancy in the turn density is as large as is the difference in radial density between the synthetic and empirical flies. A formal analysis confirms the trend (Figure S27). It shows that the difference between the radial density for empirical and synthetic flies is correlated to the increased turn density at the border of the stimulated region.

## Discussion

The four-state locomotor model implies that the behavior of a fly walking around in a small arena can be modeled as a series of discrete decisions; each decision is a choice of speed, curvature, and duration from an underlying distribution. ORN activation has two effects on behavior: First, ORN activation changes the distribution of locomotor variables. Second, because ORNs are activated only in the stimulated region, this creates an asymmetry; flies take advantage of this asymmetry by biasing their turns to direct them toward the stimulus zone.

A probabilistic model makes sense. Stimulating the fly's olfactory system alone does not provide strong directional cues; locomotion makes sensory assessment even more uncertain, as sensory delays mean that the current firing rate only reports past events at a different location. While the exact nature of the locomotor model and how sensory information, including olfactory information, affect the model parameters will depend on the species, the size of the arena, and the nature of the stimulated area, we expect that a simple probabilistic model will always capture the essence of the underlying sensorimotor transformation.

Before we discuss other implications of the data we collected and our model, we should reiterate one limitation: Because flies only enter and exit the stimulated region a few times in any experiment, we are data-limited in describing the fly's behavioral changes at high $\Delta f$ as the firing rate only changes during entry and exit. This limitation means that we likely miss some locomotor changes.

We show here that flies rely on $f$ and $\Delta f$ in a short time interval to change their motor parameters. Together $f$ and $\Delta f$ serves as adequate surrogate for integrating information across time: they can signal whether the fly is going up or down a concentration gradient or is in an area with steady stimulus concentration without any need to integrate information across time and serve as short-term memory from the time the fly enters the stimulated zone and until it leaves. Using $f$ and $\Delta f$ to modify behavioral parameters is an effective and straightforward strategy to adapt the motor program to the sensory context continuously. Take, for example, the change in speed. As the fly enters the stimulus-zone and its ORNs are increasingly more active, the fly starts moving faster. The speed decreases as it enters a constant stimulus region, and the firing rate changes are low. Together these different

effects of $f$ and $\Delta f$ on speed reflect a change in locomotor strategy: increased speed as the fly moves up the concentration gradient changes to a search behavior with a careful low-speed search of an area when the firing rate is constant. The observed change in speed—increases when there are large changes in ORN firing rate and a decrease when the firing rate is elevated but constant is consistent with our previous work[13]. Speed reduction is a hallmark of search behavior observed after an animal finds a resource or an odor that indicates the resource.

The immediate $f$ and $\Delta f$ can only signal short-term context—from the current entry into the stimulated region to exit. On a longer timescale, the sensorimotor mapping between $f$ and $\Delta f$ and locomotor parameters is altered and allows modulation of locomotion over successive encounters.

There are eight kinematic parameters that depend on $f$ and $\Delta f$; out of these eight parameters, all but the duration of sharp turns are affected by ORN activation. Out of the remaining seven, the speed of the curved walks and sharp turns are affected similarly across all genotypes and is likely a single parameter, leaving six parameters that all show different dependence on $f$ and $\Delta f$ and on the combination of ORN class activated. The distinct relationship between $f$ and $\Delta f$ and locomotor parameters is another facet of sensorimotor mapping that ensures a motor program that automatically adapts to sensory context. A decrease in curved walk speed discussed above is often—but not always—associated with an increase in curvature; this decreased speed and increased curvature is a simple, conserved strategy that allows an animal to stay close to the stimulated region; a phenomenon that is also observed in field studies[60]. Activation of even single ORN classes—*Or42b*, *Or42a*, *Or92a*, *Ir64a*, and *Or42a* and, to a lesser extent Ir75a—all result in this pattern of speed change. These results are consistent with the idea that activation of most ORN classes would result in a change in curved walk speed and curvature that initiates a local search.

Sharp turn curvature changes most as the firing rate decreases, implying that the increased sharp turn curvature is employed when the fly exits the stimulated region and wants to return. Again, a mapping between recent $f$ and $\Delta f$ and sharp turn curvature accomplishes a part of the overall motor program without the necessity of longtime temporal integration; an increase in sharp turn curvature is a conserved strategy documented in other insects[14,61] which also increases the sharpness of their turns to return to a resource patch. A previous study showed that this increase in turn amplitude is important for a fly spending more time in the stimulated region[47]. Although single ORN classes can cause some change in sharp turn curvature, activation of two ORN classes causes larger changes.

The dependence of curved walk duration on $f$ and $\Delta f$ and ORN combination is also distinct. As the fly exits the stimulated region, there is a large decrease in the curved walk duration. This change in curved walk duration means that as soon as the firing rate starts to decline, the flies stop walking and make a turn. It would make sense if this decrease in curved walk duration was strongly coupled to the increase in sharp turn curvature; this coupling is not observed. Changes in curved walk duration require the activation of many more ORN classes than the increase in sharp turn curvature, implying that they are likely to be modulated by parallel sensory circuits.

We have studied the effects of odor-guided locomotion in a small arena using both optogenetics and odors and find that odors affect multiple aspects of the fly's behavior—including every imaginable aspect of movement kinematics, turn direction—and that these effects occur through independent sensorimotor interactions[13]. In this arena, the wind cues are small or absent, and ORN activation is persistent as sometimes the fly enters the stimulated region for tens of seconds to minutes. In stark contrast, in recent work[26,62,63] in which the fly's olfactory system was stimulated with transient odor/optogenetic stimulation in the presence of constant wind, the authors reported no change in movement kinematics. In their behavior, the effect of odors

was to change walk-stop transitions and bias turn directions. Another recent study lies somewhere in the middle and finds that the presence and sensing of wind are necessary for upwind progression, but odors alone produce many changes in movement parameters[11]. These results suggest that adult flies have a rich, flexible behavioral repertoire which depends not only on the receptors that are activated but also on the temporal features of odor such as stimulus duration and encounter rate, and the presence or absence of wind. The range of behavior in nature is likely to be even more varied given that both the speed and direction of wind changes frequently[64] and the odor environment is considerably richer[65].

Another interesting difference appears to be integration time. In our study, the linear filters, even when the fly is leaving the arena and has been experiencing odor for some time, are transient and lasts less than 200 ms. This transience appears to be in contrast to the much longer integration times that describe behavior in studies in which the direction of the wind is constant. One possible explanation for this difference is that in our arena there is no clear directional source. In the absence of a source of direction, it is unclear how information sources necessary for guiding locomotion– such as location and time of an odor encounter would be integrated together. This integration likely becomes easier in the presence of wind particularly when the direction and speed of wind does not change[66].

In sum, these recent studies in the context of fly walking emphasize decades of work that has unraveled the great diversity of strategies that insects use to change their locomotion to approach the source of odors, and to stay close to them[3,5]. This wide diversity of strategies implies that all aspects of the model we present here are unlikely to be generalizable across all conditions observed by flies in nature. We anticipate the following ideas to be generalizable: (1) it is likely that the sensorimotor transformation is likely to be probabilistic irrespective of stimulus condition, (2) the analysis performed here, i.e., trying to understand how distributions of motor parameters relate to $f$ and $\Delta f$ and how these relationships evolve over multiple encounters with an odor is likely to be an adequate description of the behavior. It is possible that a more elegant model could produce a concise description of the changes in a fly's locomotion under all conditions. However, we speculate that it is more likely that behavioral flexibility will be reflected as a combination of different regions of $f$ and $\Delta f$ being employed under different stimulus conditions and different mapping between $f$ and $\Delta f$ and motor parameters under different stimulus conditions. Thus, the model proposed here can be employed to evaluate the mechanisms underlying behavioral flexibility.

How do olfactory circuits execute this seemingly complicated behavioral strategy? The circuit architecture underlying the signal propagation from the ORNs to the antennal lobe and further to the lateral horn is well-suited to perform this computation. We have illustrated these ideas by performing a limited connectomics analysis (Figure S22). The behavioral strategy outlined above requires two computations. First, the circuits should be able to compute $f-\Delta f$. Computing $f-\Delta f$ means that different higher-order neurons respond most strongly to different phases (such as the rising phase, plateau etc.). This differential response is possible because each ORN class connects to multiple second-order neurons called projection neurons (PNs). There are ~350 PNs in all; ~200 PNs are cholinergic and excitatory; ~100 PNs are GABAergic and inhibitory; neurotransmitters for the rest are uncertain[39]. The PNs are divided almost equally between uniglomerular PNs that send their dendrites to a single glomerulus and multiglomerular PNs that project to multiple glomeruli. The presence of nearly equal numbers of excitatory and inhibitory PNs implies that it would be easy to perform sign inversion in higher-order circuits. Differences in kinetics are also likely. The most well-studied PN class– excitatory uniglomerular PNs–act as differentiators due to the synaptic depression at the ORN-PN synapse[67]. It is possible that other classes of PNs in the antennal lobe have different kinetics, as the inputs

into different PNs within each glomerulus are heterogeneous. Functional heterogeneity in PN responses within a glomerulus has been observed in other insects[68,69].

Second, our data suggest that activities from two ORN classes are integrated according to different rules based on both instantaneous $f-\Delta f$ and the locomotor parameters. This integration can happen both at the level of mPNs and lateral horn. Each glomerulus makes connections with ~30-40 mPNs; the number and diversity of mPNs imply that there is enough neural circuitry to compute different rules of integration. Similarly, there is great diversity in cell types in the lateral horn. Based on this connectivity pattern, the lateral horn consists of ~500 cell types in Drosophila[40]. There are also >37 types of output neurons from the lateral horn.

As already mentioned, much of the work in odor-modulation of locomotion in flies is aimed at understanding whether a given ORN class is attractive or repulsive[48,70] or the valence of a given ORN class. In some previous studies, particular ORN classes have been found to be particularly potent attractant or repellent. In one study, activation of $Or42b$ ORN produced strong attraction[70]; however, most other studies have found $Or42b$–by itself –to be only mildly attractive[13,36]. Similarly, activation of $Ir64a$ was initially found to mediate repulsion due to acids[44]; however, other studies have found the activation of $Ir64a$ to be attractive[71]. This study supports the finding that both Or42b activation and Ir64a activation leads to mild attraction. More generally, activating a single ORN class always causes a change in some locomotor parameter and usually results in more time being spent in the vicinity of the stimulated area (Figure S1). Attraction–as defined as time spent within the stimulated area–remains small. This result–that activation of a single ORN class would result in either no attraction or mild attraction –is consistent with most work on attraction[48].

All the ORN classes activated in this study are activated by apple cider vinegar, a strong attractant. Although no single ORN class produces strong attraction, activating three ORN classes–$Ir64a$ $Ir75a$ $Or42b$ together produces a strong attraction. This change in attraction is not explained simply, i.e., no parameter is uniquely affected by these three ORN classes that are not affected by a smaller number of ORN classes. Rather, all the changes in locomotor parameters that facilitate attraction are observed when these three ORN classes are activated. These changes include slowing down inside the stimulated region, a decrease in run duration and increased sharp turn curvature as the fly exits the arena, and greater oriented turns into the stimulated area. We have not activated enough combinations of three ORN classes to assess whether this combination is particularly optimal. But, we do know that all combinations of three active ORN classes do not mediate the same attraction. When $Or42b$, $Or42a$, and $Or92a$ are activated, the attraction is not only smaller than $Ir64a$ $Ir75a$ $Or42b$ attraction; it is also less than when $Or42b$ and $Or92a$ are activated. Does this mean that $Or42a$ is a repulsive ORN class? In fact, activation of $Or42a$ by itself is mildly attractive. This combination of activated ORN classes (i.e., all three ORN classes) causes smaller attraction likely because of smaller changes in curved walk speed and duration and smaller changes in sharp turn total curvature leading to less time spent near the stimulated region. Thus, to further understand the mechanisms underlying attraction, it would be useful to compare how different combinations of three or more ORN classes affect different locomotor parameters.

An analysis of $Or42b$, $Or42a$, and $Or92a$ has another important insight. If we think of the propensity of a fly to take the optimal turn as a measure of its intention to return to the odor, then $Or42b$, $Or42a$, and $Or92a$ active flies should be very attractive as they show strong turn bias (Fig. 5) suggesting that an intent to be attracted to the stimulated region does not necessarily mean that the sensorimotor transformation necessary to achieve that intent is a given.

Taken together, this study does not support the viewpoint that each ORN has a strong ecological function. The study further strengthens arguments put forth by us[13] and others[48] that the

behavioral effect of activating a single ORN class is small, and the rules of integration are diverse and complex. The finding that activation of all *Orco*-ORNs[72]—which consists of 70% of all ORNs—results in a large attraction[36] is an example that suggests that the role of the olfactory system is not simply to signal specific ecologically relevant patterns of attraction. In this study, we show that activation of *Orco* and *Ir8a* ORNs together results in an even stronger attraction, further weakening the argument that individual ORNs have strong ecological importance.

The rules of integration between ORN classes are likely to be just as diverse as those that govern multisensory cues' integration. These rules are likely to depend on the set of ORN classes active, the state and goals of the fly, and the temporal structure of the stimulus. These rules of integration are just beginning to be understood, as exemplified by a recent study that uncovered circuit mechanisms that make $CO_2$ a repellent under some conditions and attractive in other conditions[73]. Overall, these considerations imply that the rules of integration between ORN classes and locomotor parameters that we uncover (Fig. 6) in the context of this behavior will likely change depending on the conditions above.

Apart from behavioral changes that depend on instantaneous neural response, we observe three changes that require a fly to accumulate evidence over time or remember past sensory experiences: First, there are some changes in behavior even after the ORN firing rate reaches baseline. Among these changes are kinematic changes, and the flies spend less time at the arena border. These effects represent short-term memory that lasts ~10 s and are likely mediated by dopamine-mediated circuit modifications in the fly's. Second, the kinematic changes adapt over time (Figure S15); these changes depend on the ORN class and the locomotor parameter. One prominent adaptation is that as time since the first stimulus encounter increases, flies spend more time at lower speeds and higher curvature inside the stimulated area, consistent with a more intense search. We speculate that these behavioral changes could be mediated by dopamine-mediated circuit modification in the mushroom body[74,75] and signaling to lateral horn and downstream motor circuits by mushroom body output neurons[40,76].

Finally, flies turn in the optimal direction as they exit the arena. This turning in the optimal direction is elicited by a large drop in the rate of ORN firing, but it likely implies that the fly has some spatial sense of the stimulated area and its own locomotion with respect to this area. This behavior is reminiscent of similar behavior reported in the presence of a drop of sugar[56]; the only difference is that the behavior in our arena is triggered by changes in ORN firing rate instead of purely through navigational cues. This report might be the first example of such optimal turning in Drosophila. Qualitatively similar behavior has been reported in other insects[29,30].

Much of the work on the neural basis of behavior has been performed on discrete behavior where the animal is making a binary choice or a choice between a small number of options. Discrete behaviors are self-contained—the choice is irrevocable and does not affect future choices; often, the animal has a relatively long time to decide. In this framework, the nervous system is an information processing organ[77,78]: it constructs increasingly sophisticated and abstract internal representations of the world. Many of the decisions that we make in our lives and perhaps dominate so much of our conscious mental life are also discrete. Because it makes sense, and because it is still the dominant model, a student of neuroscience would be strongly inclined to this model after picking up any neuroscience textbook[79,80].

However, many of our behaviors are not discrete. Walking to a car and making a peanut butter sandwich are all continuous behavior that requires continuous sensorimotor integration. At each instant, there is a bewildering array of choices instead of a single choice. Moreover, each choice does not result in a final outcome—reward or punishment. Current models underlying these behaviors propose a modular organization containing parallel sensorimotor loops. In many cases, each of these loops represents a solution to an aspect of an ecological problem[81,82]. In these models, there is no strict temporal hierarchy between action selection and its execution. Instead, the two occur in parallel; the environment dictates the palate of actions at any moment. Action selection occurs gradually as action execution slowly reduces the palate to a single action[83]. In these models, an internal representation of the world is not necessary.

Our data support a modular organization with parallel sensory-motor loops and provide a granular model for continuous behavior. The results are best interpreted in a control theory framework[84] in the context of multi-step behavior where the fly is endowed with a set of controls—the parameters of the locomotor model. These parameters are controlled by the state of the system—defined by $f$ and $\Delta f$—through a control policy. The mapping between $f$ and $\Delta f$ is the control policy. The goal of the control is to ensure that the fly stays close to the stimulated region and searches the stimulated region thoroughly. This mapping is constantly updated on a longer timescale as the relationship between $f$ and $\Delta f$ and different locomotor parameters is plastic; this plasticity allows both the goal and control policy to adapt as necessary.

## Methods

### Contact for reagent and resource sharing
Further information and requests for resources and reagents should be directed to and will be fulfilled by the lead contact, Dr. Vikas Bhandawat (vb468@drexel.edu).

### Experimental model and subject details
Flies were raised in sparse culture conditions consisting of 50 mL bottles of standard cornmeal media with 100–150 progeny/bottle[85]. Active dry yeast was sprinkled on each bottle after removing the parents (1–3 days) to enrich the larvae's diet. Bottles were placed in incubators set at 25 °C on a 12 h dark/12 h light cycle. 10–15 newly eclosed female flies were put on 10 mL vials of standard cornmeal media for control experiments; and on food containing all-trans-retinal (0.02% by weight retinal) for optogenetic experiments. All vials were wrapped with aluminum foil to prevent retinal degradation and to keep conditions similar in the control vials. After 3–5 days on the control food or 4–5 days on the retinal food, flies were starved by placing them in empty scintillation vials with half of a damp Kimwipe (20 µl of water/half wipe) for 15–21 h prior to experiments. Flies were anesthetized on ice prior to placing them into the behavioral arenas. All the genotypes used in this study are enumerated in Table 1.

**Behavioral experiments.** Behavioral experiments have been previously described in detail[47]. In brief, experiments were conducted in a 4 cm radius circular arena with a 1.25 cm radius central light-zone. Flies were given a 5-min light acclimation period followed which the flies were in complete darkness for another 10 min. The arena was lit with infrared light to enable tracking. The light circle was illuminated with red light (617 nm) for the last 3 min of each 6-min experiment. The fly's locomotion was recorded at 30 frames per second using an infrared video camera (Basler acA20400-90umNIR). Recorded videos were compressed to ufmf format before tracking[86]. The tracking code models flies as an oval (using the MATLAB function regionprops) to extract the body orientation and centroid positions. Head position was tracked with the criterion that the current head position should be the endpoint along the major axis that makes the smaller turn from the previous head position.

**Analysis of the distribution of the fly in the arena.** Although the optogenetic light was turned on 3-min into behavioral recording, flies started experiencing the optogenetic stimulus only after they entered the central region for the first time or "first entry". First entry was defined as the first time the fly's head enters the light zone (1.25 cm

**Table 1 | List of fly genotypes and other resources**

| Reagent or Resource | Source | Identifier |
|---|---|---|
| **Chemicals, peptides, and recombinant proteins** | | |
| All trans-Retinal | Sigma-Aldrich | R2500 |
| **Deposited data** | | |
| Raw and processed data (behavior and electrophysiology), agent models | This paper | https://doi.org/10.6084/m9.figshare.22776428 |
| **Experimental models: Organisms/strains** | | |
| *D. melanogaster*: w[1118] P{y[+t7.7] w[+mC]=20XUAS-IVS-CsChrimson.mVenus}attP18 | Bloomington Drosophila Stock Center | BDSC: 55134; FlyBase: FBst0055134 |
| *D. melanogaster*: w[1118]; P{y[+t7.7] w[+mC]=20XUAS-IVS-CsChrimson.mVenus}attP40 | Bloomington Drosophila Stock Center | BDSC: 55135; FlyBase: FBst0055135 |
| *D. melanogaster*: w[1118]; P{y[+t7.7] w[+mC]=20XUAS-IVS-CsChrimson.mVenus}attP2 | Bloomington Drosophila Stock Center | BDSC: 55136; FlyBase: FBst0055136 |
| *D. melanogaster*: w[*]; P{w[+mC]=Orco-GAL4.W}11.17; TM2/TM6B, Tb[1] | Bloomington Drosophila Stock Center | BDSC: 26818; FlyBase: FBst0026818 |
| *D. melanogaster*: w[*]; P{w[+mC]=Ir8a-GAL4.A}204.8; TM2/TM6B, Tb[1] | Bloomington Drosophila Stock Center | BDSC: 41731; FlyBase: FBst0041731 |
| *D. melanogaster*: w[*]; P{w[+mC]=Or42a-GAL4.F}48.3B | Bloomington Drosophila Stock Center | BDSC: 9970; FlyBase: FBst0009970 |
| *D. melanogaster*: w[*]; P{w[+mC]=Or42b-GAL4.F}64.3 | Bloomington Drosophila Stock Center | BDSC: 9971; FlyBase: FBst0009971 |
| *D. melanogaster*: w[*]; P{w[+mC]=Or92a-GAL4.F}62.1 | Bloomington Drosophila Stock Center | BDSC: 23139; FlyBase: FBst0023139 |
| *D. melanogaster*: w[*]; P{w[+mC]=Ir64a-GAL4.A}183.8; TM2/TM6B, Tb[1] | Bloomington Drosophila Stock Center | BDSC: 41732; FlyBase: FBst0041732 |
| *D. melanogaster*: w[*]; P{w[+mC]=Ir75a-GAL4.S}BT12.1/TM6B, Tb[1] | Bloomington Drosophila Stock Center | BDSC: 41748; FlyBase: FBst0055136 |
| **Software and algorithms** | | |
| MATLAB r2019b | MathWorks | RRID: SCR_001622 |
| any2ufmf (part of The Caltech Multiple Walking Fly Tracker) | Branson et al.[86] | http://ctrax.sourceforge.net/any2ufmf.html |
| Optogenetics arena fly tracker | Tao, Ozarkar, Bhandawat[47], This paper | https://github.com/bhandawatlab/CircularArenaTrackingCode |
| Delineation of movement states (DrosoRT) | Tao, Ozarkar, Bhandawat[47] | https://github.com/bhandawatlab/DrosoRT |
| Single sensillum recording and spike sorting GUI | This paper | https://github.com/bhandawatlab/Single-Sensillum-Spike-Sorting-GUI |
| All other software and algorithms | This paper | https://github.com/bhandawatlab/ORN-Optogenetics |

radius circle) after the light turns on. The effect of stimulation on the fly's distribution in the arena were quantified from the first entry. Behaviors were characterized using four different methods:

1. Kernel Density Estimate of spatiotemporal distributions as described above (Fig. 1c and Figure S3). Each fly's radial head position was aligned by first entry. Spatiotemporal distributions of head position were then estimated using MATLAB's ksdensity function with a Gaussian kernel.
2. Radial occupancy: The overall probability mass distribution of the average fly being a radial distance away. Since flies first enter the light zone at different time points, the weighted (by the relative amount of time after each fly's first entry) mean and standard error of the mean are calculated. A bin size of 2 mm (0.05 radial units) was utilized in generating the distribution (Figure S1 and Fig. 7).
3. Radial density of turns: This is the probability mass function of the density of head positions during the middle of sharp turns (see below). A bin size of 2 mm (0.05 radial units) was utilized in generating the distribution (Fig. 7).
4. Probability of being inside: This is the proportion of flies inside the central 1.25 cm radial location as a function of time. A 200 ms mean filter was used to smooth the trace (Figure S4).

**Electrophysiology data collection.** Single sensillum recording was performed as described previously[87]. Orco-Gal4 > UAS-Chrimson flies were held in a pipette tip using dental wax with the antenna accessible.

The antenna was positioned using glass hooks and visualized using a microscope. A single sensillum was impaled with a glass pipette filled with saline. Responses were passed through a 100x amplifier and filtered with a 5 kHz low pass Bessel filter.

Flies were illuminated using a red (617 nm) light emitting diode (LED) (Thorlabs M617L3) connected with a LED driver (Thorlabs LEDD1B) with the intensity modulated using the driver's modulation mode which follows the voltage command delivered using MATLAB. As with the behavioral experiments, the LED light was collimated (Thorlabs ACL2520U) and focused using a plano-convex lens (Thorlabs LA1433). To deliver the same range of stimulus intensity in the electrophysiology experiments as the behavioral arena, we first measured the light intensity in the behavioral arena using a photometer (Thorlabs S121C) with a 1 mm diameter precision pinhole (Thorlabs P1000D). We then placed the LED at a distance from the fly such that the range of intensity values measured from the arena maps to the driver control voltage values between 0 and 5 volts. To calibrate the light, we applied a series of voltage steps from 0 to 5 volts in intervals of 0.5 volts and measured the intensity using the pinhole. We then fit a shifted rectified linear function to map the voltage to intensity. Using these measured conversions, six 60-s behavioral positional trajectories were converted from movement paths within the behavioral arena to light intensities that the flies were stimulated with during the recording sessions (Figure S5D). The patterns were up-sampled from 30 Hz to 10 kHz for single sensillum recordings.

**Spike sorting and spike rate analysis.** Data collection and spike sorting was conducted utilizing a custom MATLAB graphical user interface (GUI). The local field potential (LFP) was found by applying a 300-ms median filter to the signal. Then the raw voltage trace was baseline subtracted by subtracting out the LFP. Spikes were identified based on valleys in voltages below −0.8 mV with a minimum time lapse between spikes of 5 ms. Spikes were sorted based on shape and size by using principal component analysis followed by k-means clustering and then manual inspection. In this study, we recorded from ab1 and ab2 sensillum. Within the ab1 sensillum, there are 4 types of neurons: ab1A-D[72,88]. Meanwhile, within the ab2 sensillum, there are 2 types of neurons: ab2A-B. These neurons are differentiated by their waveform and spike amplitude. We only considered spikes from ab1A, ab1B, and ab2A neurons. Spike rate was estimated using kernel smoothing with a 150 ms bandwidth[89].

**Filter analysis.** For each stimulus pattern, the trial with the least LFP baseline drift was baseline subtracted and set as a template. All other LFP for the stimulus pattern were linearly registered to the template LFP using a linear least squared rigid registration method[90]. The average LFP was calculated from the registered LFP traces. The LFP, firing rate, and input stimulus were down-sampled to 100 Hz for filter analysis.

The firing rate was modeled as a two-stage linear-linear filter cascade. The linear filters were fit based on previous methods[91]. Taking the first transformation from input stimulus into local field potential as an example, we calculated each linear filter as follows. First, given a $T$ time point stimulus train with $(d-1)$ time point zero padding at the beginning, we can generate a stimulus matrix $S \in \mathbb{R}^{T \times d}$ with feature length $d$. Each row that corresponds to a time point $t$ contains the stimulus train from time $t-d$ to $t$. This is equivalent to a Hankel matrix of the zero padded stimulus train. Next, we can define the response vector $r \in \mathbb{R}^{T \times 1}$ from the average local field potential from single sensillum recordings using the stimulus train. From the stimulus matrix and the response vector, we can pose the stimulus to response as a simple linear regression with a linear filter k:

$$r = Sk + \varepsilon \tag{1}$$

We solved to $k$ using Tikhonov regularization, minimizing

$$||Sk - r||^2 + ||\lambda Ik||^2 \tag{2}$$

The regularization parameter $\lambda$, which helps to prevent overfitting, was chosen based on the elbow point in the log-log plot of the regularized solution norm and the residual norm (Fig. S5B)[92]. The linear filter from LFP to firing rate was calculated using the same method.

**Quantification of kinematics.** Speed and curvature were calculated as described previously. Briefly, given the two-consecutive center of mass positions $(p_1, p_2)$, we can calculate speed as

$$s = \frac{p_2 - p_1}{\triangle t} \tag{3}$$

Then defining curvature $(k)$ as the change in the vector $(N)$ normal to the movement path, we get:

$$\alpha'(t) = -\left(\frac{dy_t}{s_t} + \frac{dy_{t+1}}{s_{t+1}}\right)\hat{i} + \left(\frac{dx_t}{s_t} + \frac{dx_{t+1}}{s_{t+1}}\right)\hat{j} \tag{4}$$

$$N = \frac{\alpha'(t)}{|\alpha'(t)|} \tag{5}$$

$$k = dN. \tag{6}$$

**Definition of locomotor states.** Each fly's movement path was classified into one of four locomotor states (stop, boundary, sharp turn, curved walk) based on a previously described method[47]. Briefly, stops were defined as when the speed was less than 0.5 mm/s. Flies were in a boundary state when the center of mass was within 1.5 mm (half a fly length) of the arena boundary. Sharp turns occurred around large peaks in curvature while curved walks did not contain large peaks in curvature. Speed and/or curvature and duration defined each locomotor state making it either 2 or 3 parameters for each state. Moving forward, duration will be lumped into kinematics for brevity. Stops were characterized by the total curvature (reorientation) and duration; boundary states by the total angle of the arc of movement around the boundary and duration; sharp turns by the total curvature, average speed, and duration; finally, curved walks were characterized by the average curvature, average speed, and duration.

**Parameterization of movement states based on kinematics.** We separated the boundary state trajectories into before and after first entry. First entry was defined as when the fly first enters the 1.25 cm light zone after the 3 min light on mark. The total angle of the arc of movement around the boundary and duration was fit to a bivariate lognormal distribution. All other kinematic distributions were generated based on ORN activity as follows: Taking sharp turns as an example, we calculated the mean firing and mean change in firing for the 200 ms interval leading up to the initiation of each sharp turn instance. Sharp turns were then separated into three categories based on the calculated ORN activity history: These were before first entry, after first entry with a non-baseline firing rate, and after first entry with baseline firing rate. For sharp turn trajectories before first entry, we independently fit each kinematic feature to a lognormal distribution.

For sharp turn trajectories with non-baseline firing rates after first entry, we generated a kinematic mapping based on neural response that is described in the next section. We fit time-dependent lognormal, beta, and exponential distributions to speed, curvature, and duration, respectively. To do this, we first implemented a sliding window of 0.5 s with a 0.3 s overlap over the time since the start of each inhibition period. We then fit the appropriate distribution (i.e., lognormal for speed) over the kinematics of the sharp turn trajectories that started within the time window. We then interpolated the distribution parameters over time using a spline function. We repeated this process for curved walks and stops. The distribution fits for after first entry with baseline firing rate and after first entry with inhibition firing rate were used in the agent-based model described in a later section.

For sharp turn trajectories with baseline firing rate after first entry, we fit the kinematic features of the first two trajectories after reaching baseline and trajectories 3 and later with two separate independent lognormal distributions. These trajectories were separated since the kinematics of the first two trajectories after reaching baseline tend to exhibit large differences from trajectories before first entry (Figure S16). However, later trajectories tend to display similar kinematics as trajectories prior to first entry. We used the Wilcoxon rank sum and estimation methods to test for significant changes in baseline firing rate kinematics from before first entry[93]. In Figure S17B, scatter plots show individual data points and corresponding error bars show mean and bootstrapped 95% confidence interval (resampled 10,000 times, bias-corrected, and accelerated). 95% confidence interval for differences between means were calculated using the same boostrapping methods.

**Estimation graphics.** We have used estimation graphics in Figures S2 and S17. Instead of calculating the p-values, estimation graphics use bootstrapping ((resampled 10,000 times, bias-corrected, and accelerated) to estimate confidence intervals for either the mean or mean differences. We show individual data points, mean, and mean differences. shows the mean, mean differences, and confidence intervals calculated using a MATLAB toolbox[93].

**Turn triggered average (Figure S17).** Sharp turns were indexed by the peak in curvature for each sharp turn trajectory. For each sharp turn, the 10-s firing rate history was defined as the turn-triggered history. Since the distribution of firing rate is highly irregular and often dominated by baseline firing rate, we only considered turns when the fly is experiencing a non-baseline firing rate. The turn triggered history for turns that occurred within 10 s of previous entry was truncated to the last entry time. The turn triggered average is the average turn triggered firing rate history across all sharp turns.

**Definition of speed, curvature, and turn probability as a function of time since entry and exit.** When flies enter and leave the light ring, they will experience a sudden increase or decrease in ORN firing rate, respectively. To determine when flies enter and leave the light ring, we first found all positive/negative peaks in firing rate greater/less than 15 spikes/s², respectively. We aligned fly tracks to entry and exit by considering +/−10 s since the peak in changes in firing rates.

The average speed and curvature were calculated across all crossing tracks for leaving and exiting. We applied a 200 ms mean filter for both transition probability and probability of being in a sharp turn state. The transition index into turns was defined by the time when a curved walk or stop state transitions into a sharp turn state. The probability of turn transition at any time after entry/exit is defined as the proportion of tracks that transition into a sharp turn. The probability of being in a sharp turn state is the proportion of tracks that are classified as being in a sharp turn state. We applied a 1-s mean filter for both transition probability and probability of being in a sharp turn state. To calculate error bars, we performed 100 resamples of 50% of the total crossing trajectories to obtain a distribution of transition and turn probability. The error bars in Figure S10 and Figure S26 are the standard deviation of the resamples.

**Generalized linear model (Fig. S10).** The turn transition probability ($\lambda$) at any time point ($t$) since crossing was fit to a generalized linear model with a logit link function in the form of:

$$logit(\lambda(t)) = \beta_0 + \beta_2 * f(t) + \beta_2 * df(t) \qquad (7)$$

Where $f(t)$ is the firing rate at time $t$ and $df(t)$ is the change in firing rate at time $t$. The same was repeated for the probability of being in a sharp turn state.

**Linear filters for speed, curvature, and turn probability as a function of time since entry and exit (Figure S26).** Derivation of linear filters for speed, curvature, and turn probability prediction was calculated using the same methodology that was used for the derivation of the filters from light intensity to ORN firing rate. Using the speed at leaving as an example, a 2-s linear filter was fit to predict the average speed based on the average firing rate response. Filters tended to predict the average speed and curvature up to 2 s after exit and entry well.

**Linear filters for speed of fly after first entry (Figure S8).** Fly-specific 2-s linear filters were fit to predict the speed of each fly after the fly's first entry based on ORN firing rate using the same methodology as other linear filter analyses.

**Linear filters to predict average kinematics during state (Figure S11).** To determine whether linear filters applied to continuous time firing rate prior to a state transition can predict the average speed and curvature of the next state trajectory, we aligned all sharp turn and curved walk trajectories by the time of state transition. Using sharp turn as an example, we calculated a 5-s linear filter to predict the average speed and total curvature of all sharp turn trajectories based on the firing rate prior to the state transition. The predicted speed and curvature of the sharp turn trajectories were a poor fit to the empirical speed and curvature.

**Neural response to kinematic mapping (or kinematic mapping).** We used a K-nearest neighbors (KNN) approach to generate kinematic mapping for each state based on neural responses. The goal is to estimate the distribution of average kinematics given ORN activity during the preceding 200 ms. Using curved walk speed as an example, we first calculated the average firing rate ($f$) and the change in firing rate ($\triangle f$) for the 200 ms window prior to the start of each curved walk trajectory. This allowed us to embed each curved walk as a point in the ($f, \triangle f$) space (Figure S5A).

To obtain the distribution of potential future curved walk speeds for a given ($f, \triangle f$) coordinate, we first divided the ($f, \triangle f$) space into grids defined by the intersection of $\triangle f$ spanning from −150 spikes/s² to 150 spikes/s² in 15 spikes/s² increments and $f$ spanning from 0 spikes/s to 55 spikes/s in 1 spikes/s increments. The range of coordinates in the ($f, \triangle f$) space was chosen to span over 99% of the possible state points within the dataset. At each coordinate in this grid, we want to use the K closest points—in terms of Euclidean distance—to compute a distribution of curved walk speeds.

Because $\triangle f$ spans over a much larger range than $f$, a normalization is necessary. In this study, we weighted the $f$ and $\triangle f$ of each point by dividing by 10 and 30, respectively (a weighting of 1 and 3 would produce identical results). These weights—selected heuristically—are roughly in line with the fact that the maximum magnitude of $\triangle f$ is ~3 times more than the maximum $f$.

Since there are locations in the ($f, \triangle f$) space where there are little to no data points, we defined a maximum Euclidean distance bound (T) that points have to fall within to be considered as part of the distribution. This means that we can represent the Euclidean distance bound (T) as an ellipse:

$$\left(\frac{f - y}{a}\right)^2 + \left(\frac{\triangle f - x}{b}\right)^2 = T^2 \qquad (8)$$

Where $a$, $b$ are the weights for $f$ and $\triangle f$, respectively, and $y$, $x$ are the coordinate locations within the ($f, \triangle f$) space. To summarize, for each $y$ and $x$ coordinate location in the ($f, \triangle f$) space, we are fitting the curved walk speed values of the K closest points—that fall within an ellipse centered at the coordinate location—to a lognormal distribution (Figure S13A). Lognormal distributions were only fit for coordinates with more than 15 trajectories (points) within bounds, as a low sample size will lead to inaccurate estimates of the underlying distribution. After fitting lognormal distributions to each coordinate location within the ($f, \triangle f$) space, we performed linear interpolation of the lognormal parameters to get the distributions of curved walk speed in any arbitrary $f$ and $\triangle f$ location in the space.

The values of K and T were selected by first calculating the standard error of the mean (SEM) over a grid search of K and T and then choosing a value near the elbow point (Figure S13C). Based on this criterion, sharp turn and curved walk kinematics were mapped to the neural response space using a K of 64 trajectories and a T of 1.5. Stop kinematics were mapped to the neural response space using a K of 64 trajectories and a T of 1.

To determine whether the kinematic mappings of flies fed on retinal were significantly different from that of control flies, we first predicted fictitious moment-by-moment firing rate profiles for control flies based on the linear filter from light intensity to firing rate. Kinematic mappings were then computed for control flies using the same method as retinal flies. At each coordinate of the ($f, \triangle f$) mapping space, we used the Kolmogorov–Smirnov test to compare the retinal distribution of movement kinematics with that of control flies not fed on retinal (Figure S14).

We used negative log-likelihood to determine if lognormal or normal distributions provide a better fit wen generating the KNN mapping. The negative log-likelihood was calculated for lognormal and normal fits at each $(f, \triangle f)$ coordinate of the KNN space and then summed to obtain the total negative log-likelihood fit over the entire KNN space for lognormal and normal fits, respectively. Lognormal distributions fit better than normal distributions for most genotypes and across most kinematic mappings due to having a higher negative log-likelihood (Figure S12).

**Adaptation in neural response to kinematic mapping (Figure S16).** To calculate the kinematic mapping in the $(f, \triangle f)$ space as a function of time since the first entry, we extended the KNN method by introducing time since the first entry $(t)$ as a third dimension. Here, we divided the $(f, \triangle f, t)$ space into grids using the same binning for the $f$ and $\triangle f$ directions as above−1 spike/s and 15 spikes/s²; in the time dimension, we used 5-s increments. In this 3-dimensional space, the Euclidean bound (T) becomes:

$$\left(\frac{f-y}{a}\right)^2 + \left(\frac{\triangle f - x}{b}\right)^2 + \left(\frac{t-z}{c}\right)^2 = T^2 \tag{9}$$

Where $a$, $b$, $c$ are the weights for $f$, $\triangle f$, and $t$, respectively, and $y, x, z$ are the coordinate locations within the $(f, \triangle f, t)$ space. To summarize, for each $y, x,$ and $z$ coordinate location in the $(f, \triangle f, t)$ space, we fit the curved walk kinematic values of the K closest points−that fall within an ellipsoid centered at the coordinate location−to a lognormal distribution (Figure S13B). In this study, we used weights of 10, 30, and 20 based on the same reasoning described in the previous section. We used the same K and T as the $(f, \triangle f)$ space kinematic mapping (see previous section).

We performed a permutation test to determine whether the $(f, \triangle f)$ space significantly changes over time since the first entry (Figure S16). Using sharp turn curvature as an example, to perform the permutation test, we randomly shuffled the time since the first entry of each sharp turn trajectory before recalculating the KNN space at each time slice. We repeated this process 100 times to calculate a distribution of KNN spaces for each time slice. We then asked for each $(f, \triangle f, t)$ coordinate, whether the empirical lognormal mean is above, below, or within the 95% confidence interval of lognormal means calculated from the shuffled KNN spaces.

**Turn optimality (Fig. 5).** Turn optimality is defined as the probability that a fly will turn in the direction that requires a smaller turn to re-orient itself towards the center of the light ring (Fig. 5b). To calculate the turn optimality, we define the current movement direction as a vector $(\vec{v}_1)$ starting at the center of mass position 200 ms prior to the state transition ($p_1$) and ending in the center of mass position at the state transition ($p_2$).

$$\vec{v} = p_2 - p_1 \tag{10}$$

Next, we defined a vector that points radially inwards toward the center of the arena from the sharp turn index ($p_2$) as:

$$\vec{u} = -p_2 \tag{11}$$

Finally, we defined a vector normal to the xy plane $(\vec{n})$. From this, we calculated the directed angle of the current direction relative to the inward vector.

$$\theta = \text{atan2}\left(\frac{(\vec{v} \times \vec{u}) \bullet \vec{n}}{\vec{v} \bullet \vec{u}}, \vec{v} \bullet \vec{u}\right) \tag{12}$$

When the directed angle is positive, the left turn is optimal; likewise, a negative directed angle indicates that rightward turning is optimal. Since a positive curvature represents a leftward turn and a negative curvature represents a rightward turn, a fly makes an optimal turn if the sign of the total sharp turn (or stops) during the next state instance is the same as the sign of the directed angle. The turn optimality ratio for a given state was defined as the total number of optimal state trajectories over the total number of state trajectories. Turn optimality was mapped to the neural response space in the method described above using a K of 64 points and a T of 1.5. Figure 5 shows the turn optimality for sharp turns. We also measured turn optimality for stops and curved walks, and these were incorporated into the model (Figure S23).

To calculate the KNN mapping of turn optimality for control flies (Figure S19), we calculated the fictitious moment-by-moment firing rate profiles by convolving the light intensity to the firing rate linear filter with the light intensity experienced by control flies. We then generated the fictitious turn optimality mapping for control flies the same way it was generated for retinal flies.

**Neural response to transition probability.** Transition probability was defined as the number of non-self-transitions from one state to another state. This was mapped to the neural response space in the method described above using a K of 128 points and a T of 1.5. After mapping to the set of locations $Y_i$ in the ORN activity space, we implemented a 5 × 5 (75 Hz/s² × 5 Hz) convolutional filter to smooth out noise due to low sample sizes, especially in regions with high $f$ and $\triangle f$. Finally, all other locations in the ORN activity space were found using linear interpolation. The transition probabilities are described below and can be generated using the accompanying code. Sharp turns always transitioned to curved walks. Curved walks largely transitioned to sharp turns except when the firing rate is low, where there is an increased transition to stops. Approximately 25 percent of stops transitioned to sharp turns and the remainder transitioned to curved walks. This transition probability did not show any noticeable trends in the neural response space and is largely similar across genotypes.

**Modeling ORN rules of summation (Fig. 6).** The goal is to understand the combinatorial effect of activating different ORN classes on changes in a sharp turn and curved walk kinematics during odor-guided locomotion. To this end, we first divided our ORN activity space into five regions (Fig. 2c and Figure S20A). Region I consists of a large positive increase in ORN firing rate defined by a threshold of 20 Hz/s². Region II consists of a high firing rate defined by a threshold of 15 Hz. Region III consists of a large negative decrease in ORN firing rate as defined by a threshold of −20 Hz/s². Region IV consists of an inhibition of firing rate as defined by a threshold of the baseline 4.7 Hz firing rate. Finally, region V consists of low ORN firing rate between baseline and 15 Hz. Within each region, the state kinematics followed an approximately lognormal distribution (Figure S20A) and are parameterized by the mean and variances of the log of the state kinematics (Figure S20A). We evaluated the rules of summation between different ORN classes in each of the five regions using a linear regression model.

Using sharp turn curvature in region 1 as an example, the mean and standard deviation of the log of the sharp turn curvature in region 1 when activating a single class of ORN is:

$$\ln(P(curvature|Region\,1)) \sim N(\mu_A, \sigma_A^2) \tag{13}$$

$$\mu_A = \mu_o + \mu_a \tag{14}$$

$$\sigma_A^2 = \sigma_o^2 + \sigma_a^2 \tag{15}$$

Simultaneous activation of a set of two classes of ORNs (A and B) is formulated as:

$$\ln(P(curvature|Region\ 1)) \sim N(\mu_{AB}, \sigma_{AB}^2) \qquad (16)$$

$$\mu_{AB} = \mu_o + \mu_a + \mu_b + \alpha_{a,b}\mu_a\mu_b \qquad (17)$$

$$\sigma_{AB}^2 = \sigma_o^2 + \sigma_a^2 + \sigma_b^2 + 2\sigma_{ab} \qquad (18)$$

Where $\mu_o$ and $\sigma_o^2$ are the genotype-specific baseline mean and standard deviation calculated from before first entry. $\mu_a$, $\mu_b$ and $\sigma_a^2$, $\sigma_b^2$ are the influence of the ORN class A and B on the mean and standard deviation of the state kinematics. Overlapping influences of ORN Class A and B are modeled using the coefficient ($\alpha_{ab}$). Finally, $\sigma_{ab}$ is the covariance between classes A and B. The model was fit using MATLAB's maximum likelihood estimation function. We fit a total of 7 ORN combinations: Orco + Ir8a, Ir64a + Ir75a, Or42b + Or92a, Or42b + Ir64a, Or42a + Or42b;Or92a, Or42b + Ir64a;Ir75a, and Ir75a + Ir64a;Or42b for sharp turn average speed, sharp turn total curvature, curved walk average speed, and curved walk average curvature. We show results for the four combinations involving Or42b (Fig. 6).

We can transform the lognormal means into the real-world state kinematics space by taking the exponent:

$$e^{\mu_{AB}} = e^{(\mu_o + \mu_a + \mu_b + \alpha_{a,b}\mu_a\mu_b)}$$
$$= e^{\mu_o}e^{\mu_a}e^{\mu_b}e^{\alpha_{a,b}\mu_a\mu_b} \qquad (19)$$

We note that the influence of each genotype ($e^{\mu_a}$, $e^{\mu_b}$) and the full interaction term ($e^{\alpha_{a,b}\mu_a\mu_b}$) each act as a multiplier to influence the real-world state kinematics. We can define these terms as the gain in kinematics due to each genotype and their interactions, respectively.

$$gain_a = e^{\mu_a} \qquad (20)$$

$$gain_b = e^{\mu_b} \qquad (21)$$

$$gain_{a,b} = e^{\alpha_{a,b}\mu_a\mu_b} \qquad (22)$$

A gain of less than 1 indicates a reduction in kinematics (e.g., decrease in speed), while a gain of greater than 1 indicates an increase in kinematics. Here, we set a threshold of 0.1 to indicate whether the gain caused by a genotype or interactions between genotypes leads to a notable change in kinematics. This means that:

$$f(x) = \begin{cases} increase, gain > 1.1 \\ decrease, gain < 0.9 \\ no\ change, o/w \end{cases} \qquad (23)$$

**Definition of synergy and antagonism, dominance, and other interactions.** Independent activity of single classes or groups of ORNs can cause increases ($\mu_a > 0$) or decreases ($\mu_a < 0$) in kinematics. When two separate classes or groups of ORNs are co-activated using Chrimson, $\alpha_{a,b}\mu_a\mu_b$ captures the potential effect of convergent downstream interactions that influence locomotor kinematics (Figure S21A). These interactions are defined to be synergistic if the interaction acts to enhance the individual effects of a single ORN class, and the interaction results in a notable change in kinematics (see above section). The interactions are defined to be antagonistic if the interactions act to cut back on the individual effects of single ORN class effects, and the interaction results in a notable change in kinematics (see above section).

Cases, where the individual ORN classes cause opposing effects, will result in other effects that cannot be directly classified as synergistic or antagonistic (Figure S21B). In these cases, one possibility is

that the resultant change in kinematics may be dominated by the activation of a single ORN class. For instance, if activating ORN class A causes an increase in kinematics, ORN class B causes a decrease in kinematics, and activating both ORNs together causes an increase in kinematics, then the change in kinematics will be dominated by ORN class A. Alternatively, there are cases where both ORNs do not cause a change in locomotor kinematics on their own, but activating both causes a notable increase or decrease in kinematics. These cases are labeled as other interactions. Finally, when the interaction term does not cause a notable change to kinematics, then the two ORN groups likely do not interact and will sum linearly through parallel pathways.

**Connectomics analysis.** To determine whether there are connections between ORNs and LHONs we first queried the Hemibrain connectomics database to extract all input and output connections for the ORNs of interest and all identified classes of uniglomerular PNs (uPNs) and multiglomerular PNs (mPNs)[38]. We then identified which of these PN classes made strong (>9) connections to any of the LHONs characterized in a recent study[94]. Because only the right hemisphere in Hemibrain is complete, any left hemisphere connections were excluded due to the possibility that connection values would be unreliable. Based on the connections between ORNs and uPNs/mPNs and between uPNs/ mPNs and the LHONs we characterized which LHONs receive inputs from ORNs either directly via their cognate uPNs or indirectly via mPNs that receive input directly from the ORNs or from their cognate uPNs (Figure S22).

**Agent-based model.** Virtual flies were initialized as described previously[47]: All synthetic flies were initialized to start at the center of a unit circular arena (normalized) centered at (0,0) with an initial heading direction along the positive x-axis (0 degrees). All flies are initialized to select a curved walk as the first state that it enters. Each simulation was run using 150 flies modeled as point objects for 6 min, with the center light ring turning on at the 3-min mark to match the experimental protocol. The simulations were run at 100 Hz since the stimulus to ORN firing rate filters was computed at 100 Hz. After the 3-min light-off period, the firing rate of flies was calculated using the previously derived linear filters and based on the light stimulus experience as a function of radial distance from the center. The first entry was defined as the first time the calculated ORN firing rate is above 10 Hz. Before the first entry, locomotor kinematics, turn optimality, and state transitions were calculated based on previously described distributions calculated from empirical flies before the first entry. After the first entry, state transitions, locomotor kinematics, and turn optimality were sampled based on kinematic mappings in the neural response space. When ORN activity is at zero (inhibition), the locomotor kinematics were sampled from inhibition distributions based on the time since the start of the inhibition period. Virtual flies performed sharp turns by moving straight for half of the duration of the sharp turn at the sampled speed, then turning based on the sampled curvature over the course of one-time step, and finally, moving straight for the remainder of the time step. Stops were implemented in the same manner, except that the speed is set to zero. Virtual flies performed curved walks by moving at the average speed and curvature for the duration of the curved walk. When the fly reaches within 1.5 mm (0.0375 normalized distance) of the arena boundary, the virtual fly enters the boundary state. Here, the fly moves around the arena boundary at a constant angular speed and duration sampled from the empirical before and after first-entry distributions. Flies exit out of the boundary state by reorientating towards the center of the arena and selecting a curved walk or sharp turn state. From the set of 150 virtual flies, only the flies with first entry times within the 85th percentile of empirical first entry times were kept. For Fig. 7c, border choice was implemented for curved walks by imposing an exponentially time decaying probability of state transition after reaching a $\triangle f$ threshold

of +/−15 Hz/s$^2$. For Figure S26C, border choice was implemented by using the convolving the linear filters for speed, curvature, and turn optimality to the firing rate history since the last entry ($\triangle f > 15\,\text{Hz/s}^2$) or exit ($\triangle f < -15\,\text{Hz/s}^2$). This filter was applied for up to 2 s after each entry/exit and reset after the fly entered or exited again.

**Correlation analysis for turn density and radial occupancy.** Synthetic fly radial occupancy was subtracted from empirical fly radial occupancy to obtain the difference in radial occupancy. Since the sum over radial distance for the difference in the probability mass function equals zero, the total positive difference measures the level of discrepancy between the empirical and synthetic flies (Figure S26A/C). The following is repeated for radial density of turns. Genotypes, where there is a larger positive position or turn difference, has a larger correlation between the difference in radial occupancy and turn density (Figure S26C).

### Reporting summary
Further information on research design is available in the Nature Portfolio Reporting Summary linked to this article.

## Data availability
All data has been deposited in Figshare under the following https://doi.org/10.6084/m9.figshare.22776428. Connectomics data can be accessed at https://neuprint.janelia.org/. Source data are provided with this paper.

## Code availability
The main code underlying the analysis in this paper is posted on Github and can be accessed through https://doi.org/10.5281/zenodo.8190933. All other code and software resources can be found in the resource table.

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

## Acknowledgements

We would like to acknowledge the members of Bhandawat lab for discussions and Barani Raman for carefully reading the manuscript. Siddhi Ozarkar, Nicholas Lent helped with experiments. This research was supported by RO1DC015827 (V.B.), RO1NS097881 (V.B.) and an NSF CAREER award (IOS-1652647 to V.B.), and an NIH F31NS120835-02 (L.T.).

## Author contributions

V.B.: Conceptualization, supervision, funding acquisition, writing, and analysis. L.T.: Conceptualization, experimentation, analysis, and writing. S.P.W.: Experimentation, analysis, and writing.

## Competing interests

The authors declare no competing interests.
