## [Peer Review File · Nature Communications]

REVIEWER COMMENTS

Reviewer #1 (Remarks to the Author):

In this manuscript Tao and Bhandawat use optogenetic activation of olfactory receptor neurons and single sensillum recordings to link the dynamics of ORN activity to locomotor behavior in walking adult flies. This follows several recent studies that have used optogenetic activation in both larvae and adult to try to predict either behavioral valence or more specific locomotor kinematics. In their paradigm, light is present in spot within a circular arena. They analyze both the distribution of flies, their behavioral kinematics, and its changes over time, then develop an agent based model that account for some but not all of the flies' behavior. Overall, I think the approach of relating ORN dynamics to behavioral dynamics is a fruitful direction for the field, and the manuscript makes many intriguing observations about variation across ORs, and time. A limitation of the present manuscript is that the model and observations are both rather complex, with many different conditions and forms of integration. This limits the generality of the model and its predictive value for either new stimulus combinations or neural circuit predictions. A few recent papers that are relevant to this study are not cited or addressed. Finally, the links to higher-order circuits seem rather speculative at this stage. The manuscript might be improved by systematizing the observations and attempting a more concise modeling framework. I also have some questions about the origin of the directional signal at the stimulus boundary.

Major issues:

1) Complexity of observations and model

The authors observe correctly that each ORN combination has a unique relationship to behavior, and that these relationships occur on many different timescales. However, the presentation of the data makes it difficult to glean big picture messages from these observations, and reach conclusions beyond the underlying complexity of the problem. Can the authors simplify their observations into a smaller set of key findings?

With regards to the sensori-motor transformation model, I am not sure why they have chosen to use filters to approximate the ORN response, but then switch to the relatively static f-df framework for the generation of locomotor motifs. Filters provide a natural language for describing sensitivity to both f and df with different weights and over different timescales, and can also incorporate slower-timescale change as the authors observe. I think it would be fruitful to try to compute filters linking ORN activity to some number of continuous locomotor parameters (which could be probabilistic) rather than chunking

the ORN timecourse into discrete states, as has been done in larvae (see below). This would also provide a more concise framework for evaluating linearity vs nonlinearity in ORN combinations.

2) Circuit hypotheses are largely speculative

The authors include a small amount of connectomic analysis on LHNs downstream of certain ORN channels. However, this part of the manuscript seems to me to be highly speculative at this point, given the complexity of the observations and model, as well as the known complexity of LHN circuits. Could not circuits in the MB also contribute to these locomotor changes? Or in the Cx which has recently been shown to be downstream of MB and LH through connectomics?

3) Some related attempts to link ORN dynamics to behavior are not cited or discussed, particularly in the larva (Shulze et al. 2015, Gepner et al. 2015, Hernandez-Nunez et al. 2015) where filters models have been used effectively to link ORN activity to behavior, and behavior is shown to depend on the dynamics of ORN activity as well its mean firing rate.

4) The origins of the directional signal the light boundary are not entirely clear to me. My understanding from the text is that the light is always on, so there is likely to be a directional signal present in light differences across the two antennae at the boundary, is that right? Is there also a role for an internal direction sense (e.g. EPGs) in this part of the behavior? IT would be helpful for the authors to clarify in the results text how these observations were incorporated into the agent based model.

5) a few minor concerns about the opto experiments: it is probably worth running an empty-Gal4 control as no ATR can still lead to low levels of activation (see recent Louis lab preprint). Also, when more than one OR-GAL4 is driving Chrimson have the authors done any measurements to show that expression levels are the same? If not this could affect the expected non-linearity of the responses.

6) Some parts of the Discussion seem overly long and not directly related to the main findings. I think this could be productively shortened.

Reviewer #2 (Remarks to the Author):

Summary.

In this paper the authors present data on how walking flies change their locomotor behavior in response to optogenetic activation of different individual ORNs, as well as several combinations of ORNs. It is nice to see the effort the authors made to connect the optogenetic activation stimulus to the firing frequency of the ORNs and the resulting behavior. The authors analyzed the locomotor activity in the context of four different behavioral states, and four locomotor parameters, in the context of a neural state space consisting of ORN firing frequency and its derivative. Finally, the authors develop an agent-based model, however, the model results do not match the fly behavior, casting some doubt on the choice of locomotor parameters used to analyze the behavior. The paper represents a great deal of experimental and analytical effort. However, the overall conclusions of this effort are murky and not supported by rigorous statistics. In the end, the primary take home message seems to be that activating individual ORNs results in unique locomotor patterns (i.e. probability distributions), and generally, activating larger groups of ORNs results in either antagonistic or synergistic changes in those locomotor statistics. The paper does not, however, present clear, detailed, and generalizable patterns, making it hard to know how these results relate to other ORN combinations, odorants, arenas, or larger scale behaviors.

Major concerns.

Throughout the results and discussion the wording often says something like “flies choose/select X from a distribution.” I don’t believe there is scientific evidence that flies are choosing behavior from given distributions. Instead, their behavior may be well described by certain distributions, but these are very different concepts. The text should be reworded to reflect that nuance correctly.

There are precious few statistical comparisons throughout the paper. The primary conclusions should be backed up by rigorous statistics, especially considering that some of the differences that are described appear to be quite subtly in the data. Very few measures of fly by fly variability for different responses are given. The lack of these statistical tests and measures of variance makes it hard to trust the conclusions.

Although the authors present some controls in Fig. 1S1, the controls are not shown for subsequent analyses. Controls are needed for the kinds of analyses shown in Fig. 4 and 5, as some of the changes in locomotor parameters could be the result of light artifacts (although flies are not super sensitive to red, they do exhibit behavioral responses to red light flashes). An in-depth discussion of controls is particularly important given recent results that certain genetic constructs can result in a reorganization of olfactory glomeruli (<https://www.biorxiv.org/content/10.1101/2022.06.16.496338v1>).

The agent-based models introduce a great deal of confusion and doubt. If the agent-based models do not recapitulate the distributions of where the real flies spent their time, how can we be confident in interpreting the locomotor parameters that were selected to analyze the flies' behavior in the first place? Is there a way to take what you learned from the agent based models to improve the choice of locomotor parameters in the first place so that the narrative is more consistent and linear?

There is an extraordinary amount of data presented here, much of it representing a great deal of effort. The overall take home message of the paper, however, is quite confusing. What concrete conclusions can we draw from the comparisons of all the different ORN activation experiments? The OR/IR combinations were selected because they are all activated by apple cider vinegar. The discussion, however, does not return to this thought process. Figure 6 provides a nice summary of the results, but what does it mean? What does it mean in the context of the overall behavior for there to be a synergistic increase in total curvature for some combinations, but an antagonistic increase for others? Is it possible to derive a general narrative or hypothesis from the results presented here? There is some discussion aimed at this question in lines 475-505. However, the overall conclusion is still quite confusing, because the relationship between increased/decreased curvature or speed is not obviously linked to attraction or aversion. The take home message that I got was that activating different receptors and combinations leads to different changes in locomotor patterns. It would be nice to have a more intuitive and descriptive understanding of what the results mean in a broader context related to real world ecology. If the agent based model behavior did a good job of recapitulating the fly behavior, then the model could potentially be used to address this by exploring how synthetic flies behavior would look in other environments, or how the specific changes in locomotor statistics observed could be connected more intuitively to search, attraction, aversion, exploration, etc.

Minor comments.

Fig 1: It would be helpful to see a distribution for at least one control fly.

Fig 1 / lines 105-110. The differences in the probability distributions for the two driver lines that are focused on (the ones shown in B), don't appear to be very obviously different. In order to make a compelling argument that they are different some kind of statistical analysis is needed (e.g. a Kolmogorov-smirnov test comparing cumulative distribution functions, but be careful to avoid pseudo replication). How many individual flies contributed to each panel in Fig. 1C? Figure 1S1 should include some measure of variance across individual flies for the distributions shown.

Line 117: Why is it surprising that the Orco/IR8a flies show stronger attraction? It seems to follow logically from the previous sentence.

Line 141 (and elsewhere perhaps): Do the authors mean "the center of the arena" instead of "the arena"? I interpret "the arena" to mean the whole behavioral arena, not just the light illuminated space.

Line 146 and elsewhere: $f-Df$ is confusing, since it can be interpreted as f minus Df . Instead, I suggest (f, Df) .

Fig 2: It would help with understanding the narrative to have labels associated with the regions that describe them, like, "entering illuminated area", "leaving illuminated area", etc.

Line 147: Do responses here mean the behavioral responses, or the estimated electrophysiological responses? Are the responses really that vastly different? They appear to be different, but I don't know about vastly.

Line 150: Is trajectory here meant to describe the path, or the trajectory through the (f, Df) state space? Be very explicit and consistent with the terminology throughout the paper.

Would the (f, Df) space generalize to other arenas, or more naturalistic scenarios in turbulent flow, where puffs of odors may be broken up into small filaments?

Lines 170-180, Fig. 3: It is difficult to relate the text to the figure and understand what is being described. It would help to have labels in the figure to know where to look, and to explain the parameters in the text. It took me a while to determine that the parameters are speed, curvature, duration, and angular velocity. It also took me a while to surmise that probability plot shown in Fig. 3 is the speed parameter during the curved walk state. Changing the label in the figure to "Curved walk: speed" might help.

Does the fly choose parameters for a given state, or does the agent in the model? Be more explicit (I don't think there is evidence that flies choose those parameters in that way?).

The title of the figure implies that the figure shows the relationship between (f, Df) and the distribution of locomotor parameters. However, only one (f, Df) pair and one locomotor parameter distribution (curved walk speed) is shown. The title of the figure caption should better reflect the figure content.

It is confusing to use the word parameters for two different things: first there are 2-3 parameters, then there are 10 parameters, and then there are eight parameters. To me it seems that there are four states and four locomotor parameters, and 10 combinations of state/parameter pairs, and 2 of those are being ignored from the subsequent analysis because behavior in the border state is not of interest? This should all be much clearer and precise.

Lines 178-180: where is it shown that, in a generalized sense, the (f, Df) determines the distribution from which the fly (agent?) chooses its parameters in the subsequent state? Where is it shown that a mapping from (f, Df) to the probably distribution describes the sensorimotor transformation? Include references to the appropriate figures/panels. I think perhaps these are shown in Fig. 4 and described in the following paragraph? Some rewording to make the narrative and evidence clear is necessary.

Line 183: where is it shown that this distribution is log normal? For all parameters?

Figure 4: analysis of the controls (ATR negative) is needed. For example, perhaps the slower speed in region V is an artifact from the red light flash. While it is true that flies response to red light is limited, it is not zero, so the potential for these artifacts needs to be discussed throughout the analysis, and in particular here.

Lines 205-215: Ir64a is involved in CO₂ detection, and is likely related to aversive behaviors, whereas Or42b is more likely related to attractive behaviors. I assume, and hope, that this will be discussed later in the paper, but I think it would help the reader to start discussing some of this in the results to provide some explanation for why different pairs of receptors may lead to different behaviors.

Lines 225-235, Fig. 4S3. The changes with time look quite subtle. Statistics are needed to make a claim that there are indeed real changes. If the changes are statistically significant, it would be helpful to have a graphical summary analysis that describes these changes more clearly that can be used to be more

descriptive in the text. “overall the sensorimotor transformation for different parameters evolves differently over time” is not very informative or descriptive.

Lines 245-250: Again, a more informative description of the results is needed. Observing that there are changes is neither descriptive nor informative. What are the changes? Can results be generalized in some intuitive or useful way?

Line 243: What kind of directional information? I believe there are a number of other papers that have also shown that walking flies and other insects use olfactory directional information, these papers should also be cited.

Fig 5: Labels describing which zone corresponds to exiting and entering the arena would be helpful.

Lines 250-255. Fig. 5 suggests that (generalizing) flies only turn in the optimal direction when in zone III. I think zone III corresponds to flies leaving the illuminated region? If this is correct, then my interpretation from the figure is that flies leaving the illuminated region tend to turn back towards the illuminated region. Meanwhile, flies that enter the illuminated region (zone I) tend to make suboptimal turns, ie. they turn away from the illuminated zone? This interpretation does not match up with the text. A much clearer explanation for what the flies doing is necessary.

Line 264 (and throughout): I would use the term “locomotor parameters” rather than “motor parameters”, given that locomotor parameters are what are analyzed here.

Fig. 6S2: Are the kinematics the same as the locomotor parameters? It would help with clarity and readability if the terms were consistent throughout the paper.

Line 315: what is the take home message of Fig. 6-S3, in relation to the data presented in this paper? What is the significance of Or42b connecting to five LHONs? What is the reader supposed to learn from this figure and analysis?

Figure 7, and associated text: The discrepancy between the synthetic flies and empirical flies in Fig. 7A and other panels is disconcerting, and confusing.

Line 365: It could also be that at each moment the fly decides to continue on its path, or to change its behavior. Is there hard evidence that can be cited that the fly chooses to do something for a given duration before making a new decision? References?

Line 369: Are a flies' decisions probabilistic, or is the result of their decisions well modeled by a probabilistic distribution? These are different scenarios and the nuance needs to be correctly conveyed.

Line 373: Is there evidence that a fly chooses motor parameters from a distribution, or is its locomotor behavior well approximated by a distribution? Using the terms choose, select, etc. implies far more about the behavior than I think there is scientific evidence for.

Line 375: My interpretation from Fig. 5 was that flies entering the illuminated region actually turn away from it (i.e. there are fewer than 50% optimal turns in Zone 1).

A detailed table of all the genetic lines, full genomes, sources (Bloomington IDs where appropriate) should be added to the methods.

A few very relevant references appear to be missing, all of which should be cited and discussed in the context of the data presented in this paper:

1. Sensing complementary temporal features of odor signals enhances navigation of diverse turbulent plumes. V Jayaram, N Kadakia, T Emonet. *Elife* 2022.
2. Encoding and control of orientation to airflow by a set of *Drosophila* fan-shaped body neurons. T Currier, A Matheson, K Nagel. *Elife* 2020.
3. Elementary sensory-motor transformations underlying olfactory navigation in walking fruit-flies. E Alvarez-Salvado et al. *eLife* 2018.
4. Odor motion sensing enables complex plume navigation. N Kadakia et al. *bioRxiv* 2021.

Reviewer #3 (Remarks to the Author):

Tao, Wechsler, and Bhandawat address a significant question in the field of olfaction: how the dynamics of sensory neurons guide motor outputs in freely moving animals exploring an odor landscape. To circumvent the difficulty of defining spatial odor environments, they use optogenetics of one or many olfactory neurons. An important aspect of their work is that they replicate the stimulus that freely moving animals encounter to measure the neural responses of different sets of ORNs. This approach is powerful because, in principle, it should enable considering behavioral feedback (the way in which behavior affects the statistics of future sensory inputs in a defined environment).

Very few studies have taken this approach (the preprint of Kadakia et al 2021 comes to mind), most of the literature measures either behavioral or neural dynamics but not both. While the methodology of measuring the neural activity that corresponds to a freely-moving behavior is strong, I find that the analysis that follows needs significant improvements. Below I detail the main improvements and then I include a list of other suggestions:

Main improvements:

In dynamic environments animals integrate information over time to then modulate their exploratory motor states. In the experimental design of this manuscript that integration occurs predominantly over time. While the derivative and absolute value of the sensors firing rate play a role in modulating behavior, analyzing those parameters exclusively results in an oversimplification of the problem and fails to reveal the dynamic sensorimotor transformations underlying odor-guided exploratory behaviors.

As it has been shown in studies that range from bacteria navigating chemical gradients, to *C. elegans*, *Drosophila* larvae, and zebrafish, temporal integration that follows linear or non-linear dynamics can be used to explain motor state transitions of exploratory behaviors. For example, in the biphasic linear filters that describe olfactory-guided behaviors in larval *Drosophila* (Gepner et al 2015, Hernandez-Nunez et al 2015, Schulze et al 2015), depending on the shape of the filter, the derivative and proportional gains of the sensors may contribute differently. The sensorimotor transformation is defined by the temporal relationship between sensory neuron dynamics and behavioral dynamics.

Many types of models including simple linear kernels, generalized linear models, etc would allow the authors to better explore the temporal relationship between ORN activity and behavior. Ideally the authors should build a model that truly maps ORN temporal activity onto motor transitions. An alternative that would validate their current analysis would be to show that the ORN activity and derivative in the previous few seconds of a motor transition are irrelevant and only the instantaneous firing rate and derivative determine motor output. Another test to the current analysis could be to show that the second derivative does not play a role. If those tests do not support the current model, the analysis may need to be reformulated. That reformulation would affect Figure 4 and the rest of the paper.

Other suggestions:

-The authors could further explain the reasoning behind their selection of the ORN groups they decided to co-activate.

- A clearer explanation of the selection of exploratory parameters is needed. Using dimensionality reduction to find the motor states would be ideal.

- The writing style could be more precise and concise. For example, in the abstract sentences 1 and 2 could be merged in one and the last sentence does not clearly explain the conclusion of the article.
- In Line 4 of the introduction, the authors should clarify that they are referring only to turbulent environments, their statement is not true for diffusion dominated spaces that occur in places where small terrestrial animals live.
- The sentence that starts on line 22 should be split in two, as it stands now it may confuse the reader.
- The sentence in line 96 mentions other 3 combinations that were not used. This sentence probably should be removed or the relevance of including that information clarified.
- As a personal preference I would recommend not using 3D for Figure 1, as the color of the heatmap conveys the same information as the z-axis.

REVIEWER COMMENTS

Reviewer #1 (Remarks to the Author):

In this manuscript Tao and Bhandawat use optogenetic activation of olfactory receptor neurons and single sensillum recordings to link the dynamics of ORN activity to locomotor behavior in walking adult flies. This follows several recent studies that have used optogenetic activation in both larvae and adult to try to predict either behavioral valence or more specific locomotor kinematics. In their paradigm, light is present in spot within a circular arena. They analyze both the distribution of flies, their behavioral kinematics, and its changes over time, then develop an agent based model that account for some but not all of the flies' behavior. Overall, I think the approach of relating ORN dynamics to behavioral dynamics is a fruitful direction for the field, and the manuscript makes many intriguing observations about variation across ORs, and time. A limitation of the present manuscript is that the model and observations are both rather complex, with many different conditions and forms of integration. This limits the generality of the model and its predictive value for either new stimulus combinations or neural circuit predictions. A few recent papers that are relevant to this study are not cited or addressed. Finally, the links to higher-order circuits seem rather speculative at this stage. The manuscript might be improved by systematizing the observations and attempting a more concise modeling framework. I also have some questions about the origin of the directional signal at the stimulus boundary.

We thank the reviewer for their thoughtful critiques. It also appears that the reviewer is supportive of our general approach and results.

Specific steps that we have taken to address their critiques is discussed in detail below. We would like to make two broader comments here.

First, regarding the complexity of our model: While it is possible that a simpler (i.e., analytical) model can be devised, the proposed model makes little computational demands on the nervous system, and hence in that respect, it is simple. Second, given the recent work on circuit connectivity from ORNs to projection neurons to the lateral horn which clearly shows massive integration, it is unlikely that the transformation between ORN activation and behavior will be simple. Finally, the manuscript already incorporates a lot of data, and a large-scale connectomic analysis is beyond the scope of this manuscript. The limited connectomic analysis was performed to support the idea that not only can the circuit support the form of integration we present, the form of integration we present is what is expected from such a circuit. Finally, we think that much of the evidence linking ORN activation to locomotion in adult *Drosophila* suggests that this transformation is not simple, and both the fly's behavioral algorithm and rules of integration between ORN classes change depend on both the spatial-temporal location of the stimulus, which odors are present, and whether there is wind or not.

Major issues:

1) Complexity of observations and model

The authors observe correctly that each ORN combination has a unique relationship to behavior, and that these relationships occur on many different timescales. However, the presentation of the data makes it difficult to glean big picture messages from these observations, and reach conclusions beyond the underlying complexity of the problem. Can the authors simplify their observations into a smaller set of key findings?

Although there are some studies – such as Bell and Wilson 2016 - in adult fly olfaction that present a simple model for the transformation from ORN activation to behavior, the preponderance of studies has not been able to explain these findings with a simple model. For example, the role of a given ORN class changes depending on context. For example, *Ir64a*

ORNs were found to be aversive in Ai et. al. 2010 but attractive by Wasserman and Frye 2016. Similarly, Badel et. al. 2016 found that the valence (whether it is attractive or repulsive) of a glomerulus switches according to context. Similarly, a recent comprehensive study came to the conclusion that integration across ORNs does not follow simple rules of summation (Tumkaya et. al. 2022). Thus, it is unlikely that the rules of integration in the olfactory system are simple.

Second, both the antennal lobe and lateral horn circuit (the most likely third-order center important for transformation between ORN activation and behavior) suggest that there is massive integration across different ORN classes further suggesting that the rules of integration are complex. Given that ORN also signals to the mushroom body and there are recurrent connections between the lateral horn and mushroom body, it is likely that the transformation is complex and not simple.

The contribution here is as a first step towards understanding how ORNs modulate locomotion on a moment-by-moment basis and connect this odor-modulated locomotion to the change in the distribution of the fly in the arena.

To understand ORN-behavior transformation, we believe that we have to embrace this complexity. However, we do understand the reviewer's concern that it is important to simplify the take-home messages and extend our discussion to how the system will work under a variety of stimulus conditions. We have done so in this version.

A summary of the work is that a fly's locomotion is well-described as a mapping between recent (f, Df) and locomotor parameters such as speed and curvature. This mapping depends both on early sensory processing (i.e., which PNs are recruited), and how sensory signals are interpreted by higher-order circuits in the brain. If the fly encounters a different stimulus environment such as a different spatial stimulus distribution, the change in its behavior can be due to 1) change in early sensory processing, i.e., which PNs are recruited, or due to 2) how higher brain circuits interpret these changes.

We strongly believe that the nature of the transformation is highly complex and highly flexible.

An important aspect of understanding this transformation is to record from higher-order neurons in the fly brain, an important part of our future plan.

I continued) With regards to the sensori-motor transformation model, I am not sure why they have chosen to use filters to approximate the ORN response, but then switch to the relatively static f-df framework for the generation of locomotor motifs. Filters provide a natural language for describing sensitivity to both f and df with different weights and over different timescales, and can also incorporate slower-timescale change as the authors observe. I think it would be fruitful to try to compute filters linking ORN activity to some number of continuous locomotor parameters (which could be probabilistic) rather than chunking the ORN timecourse into discrete states, as has been done in larvae (see below). This would also provide a more concise framework for evaluating linearity vs nonlinearity in ORN combinations.

There are two questions/critiques here. First, why do we choose the f-df framework? A more detailed answer is in response to Point#3. But the short answer is that the linear filter approaches did not work, in particular when the fly is inside the stimulated regions for long periods of time. We did obtain meaningful filters when the fly was leaving or entering the stimulated region. These filters show that the filter is incredibly transient. They often decay to 0 in <200 ms (Figure 7-S4, and reproduced below in response to comments by reviewer #3), thus justifying that f-df over a short time interval before a decision is made as a reasonable choice.

Implicit in the question about the filter is that if we could obtain a filter we could potentially obtain a more concise model such as the one obtained for the larvae. As much as one would like a simple model, all the evidence points to a more complex model. Most of the larvae models cited by the reviewer have only modeled turn rate. If one also models speed, curvature, stops – and we think we can't understand the system without it – the model will become more complex. We have clear evidence that these parameters are controlled independently (Jung, Hueston et al. 2015). As an example, curvature changes during sharp turn follow very different rules from curvature changes during curved walks – see Figure 4A. One potential simplification that we will consider in future studies is that speed can be a single parameter during curved walk and sharp turns as modulation of speed during curved walk and sharp turn is remarkably similar.

The second point is the use of locomotor states rather than continuous variables. Over the last few years, we have demonstrated that fly's locomotion is better described as a series of discrete choices rather than these choices being made every instant (Tao et. al. 2019, Tao et. al. 2020). We have also discussed how having a discrete vs. continuous model affects variability in behavior (Tao et. al. 2022). Essentially, a model that chooses from some probabilistic distribution at every instant will make too many choices/unit time and fail to capture variability.

There are couple of other reasons that justify discrete states. One is that the curvature during sharp turns and curved walk is quite different (Tao, Ozarkar et al. 2020). Second, is the presence of long stops which make it hard to model locomotion without states. Studies that have modeled continuous locomotor parameters have ignored stops. The paper from Kathy Nagel's lab is an example (Álvarez-Salvado, Licata et al. 2018). The presence of long stops also make filter approaches aimed at finding filters that relate stimulus/ORN firing rate to speed impossible.

In a nutshell, one of the reasons that filter approach does not work well here is that the signals are not dynamic. Filters are a natural language when two signals that depend on each other are dynamic. This is not true for our dataset as there are many instances when both the ORN firing rate (when the fly is inside for greater than 2 seconds) and velocity (when the fly is stopped) is not dynamic.

2) Circuit hypotheses are largely speculative

The authors include a small amount of connectomic analysis on LHNs downstream of certain ORN channels. However, this part of the manuscript seems to me to be highly speculative at this point, given the complexity of the observations and model, as well as the known complexity of LHN circuits. Could not circuits in the MB also contribute to these locomotor changes? Or in the Cx which has recently been shown to be downstream of MB and LH through connectomics?

As noted above, we performed connectomics analysis just to illustrate that the model we are proposing is not too complex for the circuit to execute. We have now explicitly made this point. We are also not claiming that MB and other parts of the brain do not contribute to the behavior. All of these points are now made explicitly. We feel that the manuscript already has a great amount of data and analysis and a full connectomic analysis is beyond the scope of this study. It is also part of our future plans to investigate the transformation of higher olfactory centers and behavior.

3) Some related attempts to link ORN dynamics to behavior are not cited or discussed, particularly in the larva (Shulze et al. 2015, Gepner et al. 2015, Hernandez-Nunez et al. 2015) where filters models have been used effectively to link ORN activity to behavior, and behavior is shown to depend on the dynamics of ORN activity as well its mean firing rate.

Thanks for pointing out these references. We had tried some versions of these analyses before moving to our analysis. Since receiving reviewer feedback, over the last few months, we have sought to try these approaches more rigorously. We have reported the results of these analyses in Figure 2-S4 to 2-S7. In essence, a linear filter-based approach does produce sensible filters that can be employed to describe the data but none of the previously employed approaches worked for predicting the data. The reason for the failure is two-fold. First, approaches employed by both Gepner et.al and Hernandez-Nunez et. al. are both based on reverse correlation and make assumptions about the data that are not satisfied in our case. One problem is that in our data the stimulus and firing rates do not change for long periods. Reverse correlation-based approaches will produce erroneous filters when faced with such data.

4) The origins of the directional signal the light boundary are not entirely clear to me. My understanding from the text is that the light is always on, so there is likely to be a directional signal present in light differences across the two antennae at the boundary, is that right? Is there also a role for an internal direction sense (e.g. EPGs) in this part of the behavior? IT would be helpful for the authors to clarify in the results text how these observations were incorporated into the agent based model.

Excellent point. The turn optimality suggests that the fly has a sense of its heading direction relative to the light zone. One possibility is that the fly is maybe using the comparison of ORN activity between their two antennae to determine odor directionality [Kadackia et al., Nature 2022]. In the past, we have performed experiments with flies in which one antenna is removed, and saw that flies were still able to perform the border-hugging behaviors, albeit with larger loops. This is reflected in a remarkably similar KNN space for turn optimality (see figure 5-S2). This suggests that bilateral comparison is not the only mechanism that can lead to optimal turning. Thus it is likely that flies are also using temporal comparisons – essentially the decrease in ORN activity can not only drive turning but also drive optimal turns.

We also find some evidence for optimal turns based on internal direction sense. When the ORN activity is inhibited, the turn optimality goes to chance (see figure 5-S2). When ORN activity returns to the baseline firing rate, we do see that flies make more optimal turns than not (~0.63 for *Orco*>Chrimson and ~0.59 for single antenna *Orco*>Chrimson flies). With *Orco*>Chrimson flies, it appears that they can make optimal turns no-matter where they are in the arena (see figure 5-S2).

In reference to the final point, turn optimality is built into the model in the same way that other parameters are, i.e, we sample from the KNN model (Figure 5). We show that the turn optimality of the synthetic flies match that of the empirical flies (Figure 7-S2).

5) a few minor concerns about the opto experiments: it is probably worth running an empty-Gal4 control as no ATR can still lead to low levels of activation (see recent Louis lab preprint). Also, when more than one OR-GAL4 is driving Chrimson have the authors done any measurements to show that expression levels are the same? If not this could affect the expected non-linearity of the responses.

There are two points here. Yes, there is a small activation of the ORNs even without retinal. This activation is difficult to observe in the ORNs but can be observed as a small response in the PNs (Figure 2-S3). We think that the empty-Gal4 experiment in the Louis Lab paper is a really important experiment when using optogenetics to establish neural connectivity. Here, most of our comparisons are between the fly's behavior before the light is turned on and after it is turned on. We are not sure that the empty-Gal4 experiment will aid the interpretation of these

experiments. We interpret this remark as a minor concern. We are happy to do this experiment if the reviewer sees this as a major concern.

Second, there is likely to be variability in ORN responses – both within different ORNs of the same type and across ORNs of different-types. However, these responses will be pooled at the level of the PN circuit making it much less variable from trial-to-trial and individual-to-individual. Moreover, the light intensities we use elicit a strong response in the ORNs (Figure 2-S2). As ORN responses are amplified in the PNs until the PNs saturate, we anticipate that most ORN responses are strong enough such that the PNs will be close to saturation. Thus, the integration rules being studied here reflect the integration between channels when each channel is strongly activated. We have explained these points in the text around L145-155.

6) Some parts of the Discussion seem overly long and not directly related to the main findings. I think this could be productively shortened.

We have reduced the length of the discussion by ~1000 words. It is now 2/3 of its original length.

Reviewer #2 (Remarks to the Author):

Summary.

In this paper the authors present data on how walking flies change their locomotor behavior in response to optogenetic activation of different individual ORNs, as well as several combinations of ORNs. It is nice to see the effort the authors made to connect the optogenetic activation stimulus to the firing frequency of the ORNs and the resulting behavior. The authors analyzed the locomotor activity in the context of four different behavioral states, and four locomotor parameters, in the context of a neural state space consisting of ORN firing frequency and its derivative. Finally, the authors develop an agent-based model, however, the model results do not match the fly behavior, casting some doubt on the choice of locomotor parameters used to analyze the behavior. The paper represents a great deal of experimental and analytical effort. However, the overall conclusions of this effort are murky and not supported by rigorous statistics. In the end, the primary take home message seems to be that activating individual ORNs results in unique locomotor patterns (i.e. probability distributions), and generally, activating larger groups of ORNs results in either antagonistic or synergistic changes in those locomotor statistics. The paper does not, however, present clear, detailed, and generalizable patterns, making it hard to know how these results relate to other ORN combinations, odorants, arenas, or larger scale behaviors.

We thank the reviewer for appreciating the effort it took to connect optogenetic activation to ORN firing rate to behavior. We also thank the reviewer for their thoughtful critiques. Our efforts in addressing the critiques are detailed below; here we want to address the comments regarding complexity.

We think that the transformation between ORN activation and changes in locomotion is complex. Although there are some studies - such as Bell and Wilson 2016 - in adult fly olfaction that present a simple model for the transformation from ORN activation to behavior, the preponderance of studies has not been able to explain these findings with a simple model.

For example, the role of a given ORN class changes depending on context. For example, *Ir64a* ORNs were found to be aversive in Ai et. al. 2010 but attractive in Wasserman and Frye 2016. Similarly, Badel et. al. 2016 found that valence of a glomerulus switches. Another recent comprehensive study came to the conclusion that integration across ORNs does not follow

simple rules of summation (Tumkaya et. al. 2022). Interestingly this study was performed using a behavioral paradigm similar to Bell and Wilson 2016. Thus, it is unlikely that the rules of integration in the olfactory system will be simple.

This complexity is supported by neural circuit: both the antennal lobe and lateral horn circuit suggest that there is massive integration across different ORN classes further suggesting that the rules of integration are complex.

The generalizable message that we have now tried to hammer home harder is fly's locomotion is well-described as a mapping between recent (f,Df) and locomotor parameters such as speed and curvature. This mapping depends both on early sensory processing (i.e., which PNs are recruited), and how sensory signals are interpreted by higher-order circuits in the brain. If the fly encounters a different stimulus environment such as a different spatial stimulus distribution, the change in its behavior can be due to 1) a change in early sensory processing, i.e., which PNs are recruited, or due to 2) how higher brain circuits interpret these changes.

Major concerns.

Throughout the results and discussion the wording often says something like "flies choose/select X from a distribution." I don't believe there is scientific evidence that flies are choosing behavior from given distributions. Instead, their behavior may be well described by certain distributions, but these are very different concepts. The text should be reworded to reflect that nuance correctly.

We completely agree with the reviewer. We have made this change.

There are precious few statistical comparisons throughout the paper. The primary conclusions should be backed up by rigorous statistics, especially considering that some of the differences that are described appear to be quite subtly in the data. Very few measures of fly by fly variability for different responses are given. The lack of these statistical tests and measures of variance makes it hard to trust the conclusions.

The reviewer makes a good point. We have addressed this point by adding statistical comparisons throughout

- 1) We have added error bars for the weighted standard error of the mean in Figure 1S1 and elaborated on it in the methods.
- 2) Although the number of flies was already in Figure 1-S1, we have also added the same in Figure 1C,
- 3) We have plotted the distribution equivalent to the retinal flies for the control flies as well (Figure 1S3),
- 5) We have performed Kolmogorov-Smirnov test at each (f,Df) coordinate to compare whether the control and retinal flies' kinematic parameters given that (f,Df) coordinate are likely to be drawn from the same distribution (Figure 4-S2). This was performed for time-averaged (f,Df) coordinates and only for the (f,Df) space where we have enough data to estimate both control and retinal distribution. Note that we did not perform statistical analyses to compare the effect of different ORN classes as these are the subject of the analyses in Figure 6.
- 6) We have performed a permutation test to show how the (f,Df) space relaxes over time. Note that the estimation of the KNN distribution at each time slice is not independent. Therefore, we performed a permutation test by calculating multiple instantiations of the KNN space by shuffling the time since of first entry associated with each state trajectory. We then asked

whether the non-shuffled version of the KNN space falls above, below, or within the 95% confidence interval of the shuffled versions (Figure 4-S4).

7) We have shown the turn optimality (f,Df) space for control flies. We did not perform any further statistical tests for each (f,Df) coordinate since unlike something like turning curvature which is defined as being drawn from a log-normal distribution, turn optimality is defined as the ratio of optimal/total turns.

Although the authors present some controls in Fig. 1S1, the controls are not shown for subsequent analyses. Controls are needed for the kinds of analyses shown in Fig. 4 and 5, as some of the changes in locomotor parameters could be the result of light artifacts (although flies are not super sensitive to red, they do exhibit behavioral responses to red light flashes). An in-depth discussion of controls is particularly important given recent results that certain genetic constructs can result in a reorganization of olfactory glomeruli (<https://www.biorxiv.org/content/10.1101/2022.06.16.496338v1>).

This particular paper mentioned is not relevant to this study because it deals with very specific bacteriophage integrase-directed insertion requiring manipulations where either the Gal4 or the attP40 site is homozygous. Neither condition occurs in our experiments.

However, the reviewer's point that we have not discussed controls much is a good one and we have rectified the same in this manuscript with analyses of control and more explanation of the control in the text. This change was made throughout the text.

The agent-based models introduce a great deal of confusion and doubt. If the agent-based models do not recapitulate the distributions of where the real flies spent their time, how can we be confident in interpreting the locomotor parameters that were selected to analyze the flies' behavior in the first place? Is there a way to take what you learned from the agent based models to improve the choice of locomotor parameters in the first place so that the narrative is more consistent and linear?

In a previous study, we have established that the agent-based model does describe the fly's radial distribution (Tao et.al. 2020 PloS Computational Biology), therefore the model parameters are not the problem. We have also shown that our model for the relationship between ORN firing and behavior works except for a single aspect of the behavior: as the flies exits the arena, they have a great propensity to go from the walking state to the turning state. Our model does capture this behavior through the decrease in walk duration but not to the extent seen in flies. Essentially, the degree of change in walk duration at the border is not captured by f/df based model because of sampling issue.

We had already presented clear evidence in support of the logic above: We showed that 1) the decreased density inside matches the increased density of the flies just outside; 2) That the difference between the synthetic flies and empirical flies is proportional to the difference between the empirical and synthetic flies propensity to turn at the border; 3) If we increase the radial turn density, the distribution of the flies becomes more peaked. All of this provides convincing evidence that we do have the correct parameters.

In this revision, we have added another analysis. We have used linear filters to describe the fly's behavior at the border. Using linear filters to model the fly's behavior at the border and the KNN space to model the behavior elsewhere, and shown that the synthetic flies closely model the behavior of the experimental flies (Figure 7-S4).

In sum, the fly's behavior changes dramatically and rapidly at the stimulus border. This change could not be captured by the KNN model but can be captured by linear filter approaches (Figure

7-S4). Thus, if the sole purpose is model the behavior, then a hybrid approach that uses KNN when the firing rate changes slowly and a linear filter approach when it changes rapidly will work well.

There is an extraordinary amount of data presented here, much of it representing a great deal of effort. The overall take home message of the paper, however, is quite confusing. What concrete conclusions can we draw from the comparisons of all the different ORN activation experiments? The OR/IR combinations were selected because they are all activated by apple cider vinegar. The discussion, however, does not return to this thought process. Figure 6 provides a nice summary of the results, but what does it mean? What does it mean in the context of the overall behavior for there to be a synergistic increase in total curvature for some combinations, but an antagonistic increase for others? Is it possible to derive a general narrative or hypothesis from the results presented here? There is some discussion aimed at this question in lines 475-505. However, the overall conclusion is still quite confusing, because the relationship between increased/decreased curvature or speed is not obviously linked to attraction or aversion. The take home message that I got was that activating different receptors and combinations leads to different changes in locomotor patterns. It would be nice to have a more intuitive and descriptive understanding of what the results mean in a broader context related to real world ecology. If the agent based model behavior did a good job of recapitulating the fly behavior, then the model could potentially be used to address this by exploring how synthetic flies behavior would look in other environments, or how the specific changes in locomotor statistics observed could be connected more intuitively to search, attraction, aversion, exploration, etc.

There are two major points here. First, is there an intuitive explanation for the relationships we find between individual ORN combinations activated and locomotor parameters affected. Second, is that since the agent-based model does not work fully, it is not as useful. Since we have covered the second point above, we will focus on the first point except to say that the agent-based model does work and represents a rather simple, and flexible model to describe the fly's behavior (see above for more details).

Regarding the first point, there are two viewpoints to the relationship between ORN activation and behavior. One viewpoint suggests that each olfactory receptor has some ecological function. There is certainly support for this idea. *Or42b* is activated by fruity odors and mediates attraction (Semmelhack and Wang 2009), *Ir64a* mediates repulsion to acids (Ai, Min et al. 2010), *Or71a* mediates attraction to yeast volatiles (Dweck, Ebrahim et al. 2015), *Ir92a* mediates attraction to amines (Min, Ai et al. 2013), *Or56a* mediates aversion to microbes (Stensmyr, Dweck et al. 2012), *Or19a* mediates oviposition to citrus substrates (Dweck, Ebrahim et al. 2013), *Or83c* mediates farnesol mediated attraction (Ronderos, Lin et al. 2014).

The roles of many of the receptors above have been challenged in subsequent studies. Contrary to earlier results claiming that *Or42b* can mediate strong attraction, most recent studies suggest that attraction due to a single ORN class is small (Jung, Hueston et al. 2015, Bell and Wilson 2016, Tumkaya, Burhanudin et al. 2022). Similarly, *Ir64a* can also cause attraction under some conditions (Wasserman, Salomon et al. 2013). Oviposition response to *Or19a* were not observed by others (Chin, Maguire et al. 2018), and the fact that *Or83c* mediates farnesol mediated attraction is not universally accepted (Mansourian and Stensmyr 2015).

Thus, there are few receptors whose role in ecology are universally accepted.

There is similar confusion if one tries to nail down the role of a given odor in ecology. Take for example, acetic acid, a relatively simple case as it activates a small number of ORN types. Acetic acid's role in fly's behavior is complex. It triggers oviposition (Joseph, Devineni et al. 2009) and upwind flight (Becher, Bengtsson et al. 2010) but also repulsion in walking flies. Some authors

have hypothesized that this difference in behavior could be due to activation of Ir75a resulting in attraction, and activation of Ir64a resulting in repulsion. However, as mentioned above activation of Ir64a itself has been reported to cause both attraction and repulsion depending on the assay. Thus, in the case of acetic acid, neither the transformation from odor to behavior nor from ORN to behavior is a simple one. Take an even simpler case, CO₂ is aversive in walking flies (Suh, Wong et al. 2004) but mediates positive chemotaxis in flight.

Similarly, given that Or42b and Or92a are both likely to be strongly activated by fermenting fruit (De Bruyne, Foster et al. 2001, Mathew, Martelli et al. 2013), and are highly conserved across Drosophilid (Guo and Kim 2007), it has been proposed that activation of these receptors should cause strong attraction. Perhaps even saturating attraction. Once again, this observation has not been borne out in this study as well as others who have found that activating a much larger set of ORNs is more attractive.

All of this brings us to the second viewpoint. In this viewpoint, the effect of activating a single receptor is small or none at all (Knaden, Strutz et al. 2012) (Jung, Hueston et al. 2015, Bell and Wilson 2016, Tumkaya, Burhanudin et al. 2022), rules of addition are complex (Knaden, Strutz et al. 2012) (Tumkaya, Burhanudin et al. 2022) and context-dependent (Badel, Ohta et al. 2016).

The most recent and complete study (Tumkaya, Burhanudin et al. 2022) finds that *"Rather, the ORN combinations have diverse, intermediate valence weights within the weight space. Moreover, these weights shift as the stimulus intensity increases, and appear to even swap dominance at different intensities. These features suggest complex ORN-channel interactions."*

Our study supports these ideas. While complexity of the sensorimotor transformation might not be intuitive, they are strongly supported by circuit architecture. In particular, the intermixing of glomerular channels in the lateral horn strongly suggests that the rules of integration are complex.

Following the reviewer's critique, we have now discussed in more detail what the rules of integration we observe might mean.

Minor comments.

Fig 1: It would be helpful to see a distribution for at least one control fly.

We have now included this for all flies. See Supplementary Figure 1S3

Fig 1 / lines 105-110. The differences in the probability distributions for the two driver lines that are focused on (the ones shown in B), don't appear to be very obviously different. In order to make a compelling argument that they are different some kind of statistical analysis is needed (e.g. a Kolmogorov-smirnov test comparing cumulative distribution functions, but be careful to avoid pseudo replication). How many individual flies contributed to each panel in Fig. 1C? Figure 1S1 should include some measure of variance across individual flies for the distributions shown.

We have now made this comparison. Indeed, the distributions are statistically different. The numbers are/were included in Supplementary Figure 1S1.

Line 117: Why is it surprising that the Orco/IR8a flies show stronger attraction? It seems to follow logically from the previous sentence.

We have included an explanation for why it is surprising. A naïve assumption would be that over stimulating the olfactory system would be highly aversive. Also, if we follow conventional logic – some ORNs are attractive, others are repulsive so acting all ORN classes together would lead to some cancellation. Overall, the result that activating all Orco ORNs itself is surprising; that we get increased attraction when we activate even more ORNs is even more surprising.

Line 141 (and elsewhere perhaps): Do the authors mean "the center of the arena" instead of "the arena"? I interpret "the arena" to mean the whole behavioral arena, not just the light illuminated space.

Apologies for the oversight. Yes, we do mean return to the stimulated area, and we have changed this.

Line 146 and elsewhere: $f-Df$ is confusing, since it can be interpreted as f minus Df . Instead, I suggest (f, Df) .

This is a great suggestion. We are using the suggested convention.

Fig 2: It would help with understanding the narrative to have labels associated with the regions that describe them, like, "entering illuminated area", "leaving illuminated area", etc.

This is a great suggestion. We have added these labels in Figure 2.

Line 147: Do responses here mean the behavioral responses, or the estimated electrophysiological responses? Are the responses really that vastly different? They appear to be different, but I don't know about vastly.

We have removed the word "vastly"

Line 150: Is trajectory here meant to describe the path, or the trajectory through the (f, Df) state space? Be very explicit and consistent with the terminology throughout the paper.

Good point. This refers to trajectory through the (f,df) space. We have changed our writing around this point.

Would the (f, Df) space generalize to other arenas, or more naturalistic scenarios in turbulent flow, where puffs of odors may be broken up into small filaments?

Yes. We have discussed this now.

Lines 170-180, Fig. 3: It is difficult to relate the text to the figure and understand what is being described. It would help to have labels in the figure to know where to look, and to explain the parameters in the text. It took me a while to determine that the parameters are speed, curvature, duration, and angular velocity. It also took me a while to surmise that probability plot shown in Fig. 3 is the speed parameter during the curved walk state. Changing the label in the figure to "Curved walk: speed" might help.

Good point. We have amended the figure and writing to explain this figure more clearly

Does the fly choose parameters for a given state, or does the agent in the model? Be more explicit (I don't think there is evidence that flies choose those parameters in that way?).

Good point. We have made this change

The title of the figure implies that the figure shows the relationship between (f,Df) and the distribution of locomotor parameters. However, only one (f,Df) pair and one locomotor parameter distribution (curved walk speed) is shown. The title of the figure caption should better reflect the figure content.

We disagree on this point. The title says "framework". This figure describes our analytical framework.

It is confusing to use the word parameters for two different things: first there are 2-3 parameters, then there are 10 parameters, and then there are eight parameters. To me it seems that there are four states and four locomotor parameters, and 10 combinations of state/parameter pairs, and 2 of those are being ignored from the subsequent analysis because behavior in the border state is not of interest? This should all be much clearer and precise.

We have tweaked our description here. Hopefully it is clearer.

Lines 178-180: where is it shown that, in a generalized sense, the (f,Df) determines the distribution from which the fly (agent?) chooses its parameters in the subsequent state? Where is it shown that a mapping from (f,Df) to the probably distribution describes the sensorimotor transformation? Include references to the appropriate figures/panels. I think perhaps these are shown in Fig. 4 and described in the following paragraph? Some rewording to make the narrative and evidence clear is necessary.

We have tweaked our description here to better articulate that figure 3 is the analytical framework and figure 4 are the results.

Line 183: where is it shown that this distribution is log normal? For all parameters?

Yes, good point. It is lognormal for most parameters. This is now shown in Figure 3-S1

Figure 4: analysis of the controls (ATR negative) is needed. For example, perhaps the slower speed in region V is an artifact from the red light flash. While it is true that flies response to red light is limited, it is not zero, so the potential for these artifacts needs to be discussed throughout the analysis, and in particular here.

Yes. We have included an analysis for the control. The slower speed in region V is not an artifact and reflects a real effect.

Lines 205-215: Ir64a is involved in CO2 detection, and is likely related to aversive behaviors, whereas Or42b is more likely related to attractive behaviors. I assume, and hope, that this will be discussed later in the paper, but I think it would help the reader to start discussing some of this in the results to provide some explanation for why different pairs of receptors may lead to different behaviors.

Much of the early work in relating ORN activation to behavior seemed to confer specific and potent functions to ORNs. Some researchers still hold this view. However, much of the recent work linking ORN activation to behavior suggests that 1) contribution of individual ORN class to behavior is small and flexible. In the specific case of Ir64a – the first study (Ai. et. al 2010) did suggest that IR64a plays a role in aversive behavior. However, other studies have shown a role for Ir64a in attraction rather than aversion. One example is Bell and Wilson 2016 – which shows that Ir64a mediates attraction in a walking arena. Another example is Wasserman 2013.

Note also that Ir64a is activated by apple cider vinegar, a known attractant.

In terms of relating odors to ORNs to behavior, these relationships are not simple. A given odor can be "attractive" or "repulsive". As an example, acetic acid is attractive as it triggers oviposition (Joseph et. al. 2009) and upwind flight (Becher et. al. 2010) but also repulsion in walking flies. Moreover, most odors do not activate a single ORN class.

In sum, links between ecological role of an odor and behavior elicited by ORNs activated by those odors are complex expect for a few specialist odors and the ORN classes they activate.

There are some important hints on how the system functions in the behavior downstream of three active ORN classes: When Or42a is activated in addition to Or42b and Or92a, it leads to less

attraction whereas when Ir64a, Ir75a and Or42b are activated, it leads to much stronger attraction. Note that all of these ORN classes are activated by apple cider vinegar. Perhaps the rules of integration would be clearer if one activated multiple combinations of three ORN classes.

Lines 225-235, Fig. 4S3. The changes with time look quite subtle. Statistics are needed to make a claim that there are indeed real changes. If the changes are statistically significant, it would be helpful to have a graphical summary analysis that describes these changes more clearly that can be used to be more descriptive in the text. "overall the sensorimotor transformation for different parameters evolves differently over time" is not very informative or descriptive.

This is a great point. We have now performed statistical analysis.

Lines 245-250: Again, a more informative description of the results is needed. Observing that there are changes is neither descriptive nor informative. What are the changes? Can results be generalized in some intuitive or useful way?

We have rewritten this section.

Line 243: What kind of directional information? I believe there are a number of other papers that have also shown that walking flies and other insects use olfactory directional information, these papers should also be cited.

Thank you for pointing this out and we apologize not discussing this in detail. We have added citations to these papers and added some discussion about directionality.

Fig 5: Labels describing which zone corresponds to exiting and entering the arena would be helpful.

We have added these labels.

Lines 250-255. Fig. 5 suggests that (generalizing) flies only turn in the optimal direction when in zone III. I think zone III corresponds to flies leaving the illuminated region? If this is correct, then my interpretation from the figure is that flies leaving the illuminated region tend to turn back towards the illuminated region. Meanwhile, flies that enter the illuminated region (zone I) tend to make suboptimal turns, ie. they turn away from the illuminated zone? This interpretation does not match up with the text. A much clearer explanation for what the flies doing is necessary.

Thank you for pointing this out. You are correct in noticing that the analysis shows that when flies leave, they tend to make optimal turns inwards and when they re-enter, they will make un-optimal turns outwards. This will result in a border weaving pattern that we often see in fly tracks. We have clarified this point in the text.

Line 264 (and throughout): I would use the term "locomotor parameters" rather than "motor parameters", given that locomotor parameters are what are analyzed here.

Good point. We have done so now.

Fig. 6S2: Are the kinematics the same as the locomotor parameters? It would help with clarity and readability if the terms were consistent throughout the paper.

Yes. We now use locomotor parameters or words referring to parameter such as "speed"

Line 315: what is the take home message of Fig. 6-S3, in relation to the data presented in this paper? What is the significance of Or42b connecting to five LHONs? What is the reader supposed to learn from this figure and analysis?

Good point. We did not make this clear in our previous version. We have explained this much better now. The point is to show that even if we just look at the connection from antennal lobe to

lateral horn, the circuit suggests that integration between different ORN classes is unlikely to be simple.

Figure 7, and associated text: The discrepancy between the synthetic flies and empirical flies in Fig. 7A and other panels is disconcerting, and confusing.

Please see our response under the major points.

Line 365: It could also be that at each moment the fly decides to continue on its path, or to change its behavior. Is there hard evidence that can be cited that the fly chooses to do something for a given duration before making a new decision? References?

This is similar to your point earlier which is a good one. We agree that language that describes a fly as making certain decisions is incorrect. However, there is evidence from our work that there is persistence in behavior (Tao, Ozarkar et al. 2019, Tao and Bhandawat 2022).

Line 369: Are a flies' decisions probabilistic, or is the result of their decisions well modeled by a probabilistic distribution? These are different scenarios and the nuance needs to be correctly conveyed.

This is similar to your point earlier. We have stated these findings as a fly's behavior being modeled by a process that has certain characteristics.

Line 373: Is there evidence that a fly chooses motor parameters from a distribution, or is its locomotor behavior well approximated by a distribution? Using the terms choose, select, etc. implies far more about the behavior than I think there is scientific evidence for.

This is similar to your point earlier. We have stated these findings as a fly's behavior being modeled by a process that has certain characteristics.

A detailed table of all the genetic lines, full genomes, sources (Bloomington IDs where appropriate) should be added to the methods.

Table for genetic lines has been added.

A few very relevant references appear to be missing, all of which should be cited and discussed in the context of the data presented in this paper:

- 1. Sensing complementary temporal features of odor signals enhances navigation of diverse turbulent plumes. V Jayaram, N Kadakia, T Emonet. Elife 2022.*
- 2. Encoding and control of orientation to airflow by a set of Drosophila fan-shaped body neurons. T Currier, A Matheson, K Nagel. Elife 2020.*
- 3. Elementary sensory-motor transformations underlying olfactory navigation in walking fruit-flies. E Alvarez-Salvado et al. eLife 2018.*
- 4. Odor motion sensing enables complex plume navigation. N Kadakia et al. bioRxiv 2021.*

This is a good point. We had ignored these studies as they focus on a single odor. But looking at them allows us to make the point that behaviors elicited by odors change dramatically with context. These references have been added.

Reviewer #3 (Remarks to the Author):

Tao, Wechsler, and Bhandawat address a significant question in the field of olfaction: how the dynamics of sensory neurons guide motor outputs in freely moving animals exploring an odor landscape. To circumvent the difficulty of

defining spatial odor environments, they use optogenetics of one or many olfactory neurons. An important aspect of their work is that they replicate the stimulus that freely animals encounter to measure the neural responses of different sets of ORNs. This approach is powerful because, in principle, it should enable considering behavioral feedback (the way in which behavior affects the statistics of future sensory inputs in a defined environment). Very few studies have taken this approach (the preprint of Kadakia et al 2021 comes to mind), most of the literature measures either behavioral or neural dynamics but not both. While the methodology of measuring the neural activity that corresponds to a freely-moving behavior is strong, I find that the analysis that follows needs significant improvements. Below I detail the main improvements and then I include a list of other suggestions:

Thank you for appreciating our experimental approach and for the incisive critiques below.

Main improvements:

In dynamic environments animals integrate information over time to then modulate their exploratory motor states. In the experimental design of this manuscript that integration occurs predominantly over time. While the derivative and absolute value of the sensors firing rate play a role in modulating behavior, analyzing those parameters exclusively results in an oversimplification of the problem and fails to reveal the dynamic sensorimotor transformations underlying odor-guided exploratory behaviors.

As it has been shown in studies that range from bacteria navigating chemical gradients, to C. elegans, Drosophila larvae, and zebrafish, temporal integration that follows linear or non-linear dynamics can be used to explain motor state transitions of exploratory behaviors. For example, in the biphasic linear filters that describe olfactory-guided behaviors in larval Drosophila (Gepner et al 2015, Hernandez-Nunez et al 2015, Schulze et al 2015), depending on the shape of the filter, the derivative and proportional gains of the sensors may contribute differently. The sensorimotor transformation is defined by the temporal relationship between sensory neuron dynamics and behavioral dynamics.

Many types of models including simple linear kernels, generalized linear models, etc would allow the authors to better explore the temporal relationship between ORN activity and behavior. Ideally the authors should build a model that truly maps ORN temporal activity onto motor transitions. An alternative that would validate their current analysis would be to show that the ORN activity and derivative in the previous few seconds of a motor transition are irrelevant and only the instantaneous firing rate and derivative determine motor output. Another test to the current analysis could be to show that the second derivative does not play a role. If those tests do not support the current model, the analysis may need to be reformulated. That reformulation would affect Figure 4 and the rest of the paper.

This is an excellent point that we had spent a lot of time thinking about this issue before our first submission and we have once again done the same after reading the reviewer's critiques.

We had tried some versions of these analyses before moving to our analysis. Since receiving reviewer feedback, over the last few months, we have sought to try these approaches more rigorously. We have reported the results of these analyses in Figure 2-S4 to 2-S7. In essence, a linear filter-based approach does produce sensible filters that can be employed to describe the data but none of previously employed approaches worked for predicting the data. The reason for the failure is two-fold. First, approaches employed by both Gepner et.al and Hernandez-Nunez et. al. is both based on reverse correlation and make assumptions about the data that are not satisfied in our case. One problem is that in our data the stimulus and firing rates do not change for long periods. Reverse-correlation based approaches will produce erroneous filters when faced with such data.

Linear filter approaches do describe the behavior when the fly enters or leaves the arena (Figure

7-S4, and reproduced below). These filters are incredibly transient. They often decay to 0 in <200 ms. This transience lends support to our use of (f,df) in a short time interval.

Linear filters could be obtained for the time periods when the fly is leaving or entering the arena and show that the filters are incredibly transient. These plots are for when fly is leaving the arena and filters for speed, curvature and turn transition all reach 0 in about 100 millisecond implying that the fly can indeed "ignore" prior information. These filters are reproduced from Figure 7-S4 with an expanded time scale.

Other suggestions:

-The authors could further explain the reasoning behind their selection of the ORN groups they decided to co-activate.

Thanks for the suggestion. We have done so now.

- A clearer explanation of the selection of exploratory parameters is needed. Using dimensionality reduction to find the motor states would be ideal.

We apologize but we do not understand this suggestion

- The writing style could be more precise and concise. For example, in the abstract sentences 1 and 2 could be merged in one and the last sentence does not clearly explain the conclusion of the article.

We have tried to make the text more concise

- In Line 4 of the introduction, the authors should clarify that they are referring only to turbulent environments, their statement is not true for diffusion dominated spaces that occur in places where small terrestrial animals live.

Interesting. We would very much appreciate an actual example. However, we have noted this in the text.

- The sentence that starts on line 22 should be split in two, as it stands now it may confuse the reader.

This sentence is changed.

- The sentence in line 96 mentions other 3 combinations that were not used. This sentence probably should be removed or the relevance of including that information clarified.

We are saying that three other combinations were used. We have rewritten this to make it clearer.

- As a personal preference I would recommend not using 3D for Figure 1, as the color of the heatmap conveys the same information as the z-axis.

Thank you for the suggestion. We recognize that 3D may seem redundant. However, the purpose of the z axis and the colormap is slightly different. The z-axis is shared among all genotype subplots. This allows the reader to compare across genotypes. Meanwhile, the colormap is genotype specific. This allows the reader to visually compare density at different spatial-temporal locations within a genotype. We have now mentioned this in the legend to Figure 1.

REFERENCES

- Ai, M., S. Min, Y. Grosjean, C. Leblanc, R. Bell, R. Benton and G. S. Suh (2010). "Acid sensing by the *Drosophila* olfactory system." Nature **468**(7324): 691-695.
- Álvarez-Salvado, E., A. M. Licata, E. G. Connor, M. K. McHugh, B. M. King, N. Stavropoulos, J. D. Victor, J. P. Crimaldi and K. I. Nagel (2018). "Elementary sensory-motor transformations underlying olfactory navigation in walking fruit-flies." Elife **7**: e37815.
- Badel, L., K. Ohta, Y. Tsuchimoto and H. Kazama (2016). "Decoding of context-dependent olfactory behavior in *Drosophila*." Neuron **91**(1): 155-167.
- Becher, P. G., M. Bengtsson, B. S. Hansson and P. Witzgall (2010). "Flying the fly: long-range flight behavior of *Drosophila melanogaster* to attractive odors." Journal of chemical ecology **36**: 599-607.
- Bell, J. S. and R. I. Wilson (2016). "Behavior reveals selective summation and max pooling among olfactory processing channels." Neuron **91**(2): 425-438.
- Chin, S. G., S. E. Maguire, P. Huovalia, G. S. Jefferis and C. J. Potter (2018). "Olfactory neurons and brain centers directing oviposition decisions in *Drosophila*." Cell reports **24**(6): 1667-1678.
- De Bruyne, M., K. Foster and J. R. Carlson (2001). "Odor coding in the *Drosophila* antenna." Neuron **30**(2): 537-552.
- Dweck, H. K., S. A. Ebrahim, A. Farhan, B. S. Hansson and M. C. Stensmyr (2015). "Olfactory proxy detection of dietary antioxidants in *Drosophila*." Current Biology **25**(4): 455-466.
- Dweck, H. K., S. A. Ebrahim, S. Kromann, D. Bown, Y. Hillbur, S. Sachse, B. S. Hansson and M. C. Stensmyr (2013). "Olfactory preference for egg laying on citrus substrates in *Drosophila*." Current Biology **23**(24): 2472-2480.
- Guo, S. and J. Kim (2007). "Molecular evolution of *Drosophila* odorant receptor genes." Molecular biology and evolution **24**(5): 1198-1207.
- Joseph, R. M., A. V. Devineni, I. F. King and U. Heberlein (2009). "Oviposition preference for and positional avoidance of acetic acid provide a model for competing behavioral drives in *Drosophila*." Proceedings of the National Academy of Sciences **106**(27): 11352-11357.
- Jung, S.-H., C. Hueston and V. Bhandawat (2015). "Odor-identity dependent motor programs underlie behavioral responses to odors." Elife **4**: e11092.
- Knaden, M., A. Strutz, J. Ahsan, S. Sachse and B. S. Hansson (2012). "Spatial representation of odorant valence in an insect brain." Cell reports **1**(4): 392-399.
- Mansourian, S. and M. C. Stensmyr (2015). "The chemical ecology of the fly." Current opinion in neurobiology **34**: 95-102.
- Mathew, D., C. Martelli, E. Kelley-Swift, C. Brusalis, M. Gershow, A. D. Samuel, T. Emonet and J. R. Carlson (2013). "Functional diversity among sensory receptors in a *Drosophila* olfactory circuit." Proceedings of the National Academy of Sciences **110**(23): E2134-E2143.
- Min, S., M. Ai, S. A. Shin and G. S. Suh (2013). "Dedicated olfactory neurons mediating attraction behavior to ammonia and amines in *Drosophila*." Proceedings of the National Academy of Sciences **110**(14): E1321-E1329.
- Ronderos, D. S., C.-C. Lin, C. J. Potter and D. P. Smith (2014). "Farnesol-detecting olfactory neurons in *Drosophila*." Journal of Neuroscience **34**(11): 3959-3968.

- Semmelhack, J. L. and J. W. Wang (2009). "Select *Drosophila* glomeruli mediate innate olfactory attraction and aversion." Nature **459**(7244): 218-223.
- Stensmyr, M. C., H. K. Dweck, A. Farhan, I. Ibba, A. Strutz, L. Mukunda, J. Linz, V. Grabe, K. Steck and S. Lavista-Llanos (2012). "A conserved dedicated olfactory circuit for detecting harmful microbes in *Drosophila*." Cell **151**(6): 1345-1357.
- Suh, G. S., A. M. Wong, A. C. Hergarden, J. W. Wang, A. F. Simon, S. Benzer, R. Axel and D. J. Anderson (2004). "A single population of olfactory sensory neurons mediates an innate avoidance behaviour in *Drosophila*." Nature **431**(7010): 854-859.
- Tao, L. and V. Bhandawat (2022). "Mechanisms of variability underlying odor-guided locomotion." Frontiers in Behavioral Neuroscience: 118.
- Tao, L., S. Ozarkar, J. M. Beck and V. Bhandawat (2019). "Statistical structure of locomotion and its modulation by odors." Elife **8**: e41235.
- Tao, L., S. Ozarkar and V. Bhandawat (2020). "Mechanisms underlying attraction to odors in walking *Drosophila*." PLoS computational biology **16**(3): e1007718.
- Tumkaya, T., S. Burhanudin, A. Khalilnezhad, J. Stewart, H. Choi and A. Claridge-Chang (2022). "Most primary olfactory neurons have individually neutral effects on behavior." Elife **11**: e71238.
- Wasserman, S., A. Salomon and M. A. Frye (2013). "*Drosophila* tracks carbon dioxide in flight." Current Biology **23**(4): 301-306.

REVIEWER COMMENTS

Reviewer #1 (Remarks to the Author):

In this revised manuscript, Tao and colleagues address some but not all of my concerns with the original submission. Overall I find this to be a valuable descriptive account of the relationship between ORN firing rates and behavior in a specific setting. In particular, the findings that different locomotor parameters are differentially modulated by ORN firing rate over time builds on previous studies from this group (Jung et al), and the finding that flies preserve some spatial memory of the location of the odor on leaving the arena— even in the absence of bi-antennal comparisons—adds to our increasing understanding of the complexities and computations associated with odor-guided search behavior. However, I still feel that the modeling efforts here are so specific and complex as to provide little generalizing power to other contexts or paradigms, and that the circuit work on the LH is extremely preliminary and should really be treated as Discussion points rather than Results. My broad suggestion would be to treat the model(s) as an analysis of the relationships between ORN firing patterns and locomotion, rather than as a model per se, and to use the final modeling exercise at the end as a test to see how much explanatory power these analyses have (answer: some but not complete). I think the LH connectomics should be removed and saved for another manuscript that explores the roles of these neurons and connections experimentally.

Minor comments:

line 122: text mentions IR8a in Fig 1C but I do not see this in the figure.

line 134: “others have observed” should have citations

line 170-174: the fact that reasonable filters cannot be fit to the data using reverse correlation because of limitations of the dataset does not mean that a filter-based description of the behavior could provide predictive power. Overall I don't think this section adds to the manuscript as filter and glm-based approaches have shown predictive power in other behavioral contexts.

line 176: many GLMs incorporate time history, e.g.

line 232: data limitations. I think this should be addressed in the Discussion as well. To what extent do failures of the model(s) or differences between conditions reflect data limitations versus more complexity?

line 264: KNN: please define

line 286: weaving behavior: this is quite interesting and could be developed more.

line 323:329: these seem like Discussion points, not Results.

line 447-451: "exact nature of the model" this specificity is what leads me to think that the current model will not be generally applicable, and might be better framed as an analysis rather than a model.

line 608-610: short-term memory and dopamine. I think it would be helpful to break these two points up here. First the authors show evidence for short-term memory which is exciting and important. Second they speculate that it is due to dopamine which is a pure speculation based on the current data.

line 617: spatial sense of the stimulated arena. I think this is the first published evidence for this and the authors could draw more attention to it.

Reviewer #2 (Remarks to the Author):

I appreciate the effort the authors have done towards addressing my concerns.

Reviewer #3 (Remarks to the Author):

The authors have carefully considered the alternative modeling options and appropriately substantiated their choices for the main text. I consider it reasonable to publish this upgraded version of the paper.

Response to reviewer comments

Reviewer #1 (Remarks to the Author):

In this revised manuscript, Tao and colleagues address some but not all of my concerns with the original submission. Overall I find this to be a valuable descriptive account of the relationship between ORN firing rates and behavior in a specific setting. In particular, the findings that different locomotor parameters are differentially modulated by ORN firing rate over time builds on previous studies from this group (Jung et al), and the finding that flies preserve some spatial memory of the location of the odor on leaving the arena— even in the absence of bi-antennal comparisons—adds to our increasing understanding of the complexities and computations associated with odor-guided search behavior. However, I still feel that the modeling efforts here are so specific and complex as to provide little generalizing power to other contexts or paradigms, and that the circuit work on the LH is extremely preliminary and should really be treated as Discussion points rather than Results. My broad suggestion would be to treat the model(s) as an analysis of the relationships between ORN firing patterns and locomotion, rather than as a model per se, and to use the final modeling exercise at the end as a test to see how much explanatory power these analyses have (answer: some but not complete). I think the LH connectomics should be removed and saved for another manuscript that explores the roles of these neurons and connections experimentally.

We thank the reviewer for appreciating the value of this work and for their suggestions. The reviewer has two main critiques which we have addressed in the following way:

1)The model is not generalizable: We were not able to take the suggestion of not introducing the model early on in the paper (in Figure 3), because describing the model before we discuss the effect of ORNs on motor parameters is necessary as it introduces the parameters being assessed. However, we have added additional sentences in the Discussion to include the reviewer’s viewpoint.

We sense that the reviewer’s viewpoint is that there is a simple, generalizable model that would describe the sensorimotor transformation between ORN activation and locomotion under all/many conditions. Given the known diversity of the responses of flies to odors, we think that a simple, generalizable model is unlikely. Rather, we think that the work performed here provides an analytical framework for assessing how a fly’s locomotor responses change under different experimental conditions. We have now further discussed these ideas by adding them to our Discussion. See the section on “***Mechanisms for odor modulation of locomotion are highly flexible and depend on context and sensory cues***”

2)LH connectomics: Once again, we cannot go all the way in removing the LH connectomics data as we find that even the limited connectomics is helpful for readers. Connectomics is particularly helpful for all readers who do not work in the systems neuroscience of fly olfaction because it gives them some reassurance that there are circuits in *Drosophila* that could perform computations outlined in this manuscript. However, we have added additional sentences to indicate further that the connectomics are meant to illustrate that fly olfactory circuits can subserve the computations outlined here. We do agree with the reviewer that comprehensive connectomics belongs in another study in which we study higher-order olfactory circuits.

Minor comments:

line 122: text mentions IR8a in Fig 1C but I do not see this in the figure.

It is there in that figure. It is in the middle panel in the last row. Perhaps it was not clear that it is part of Figure 1C because some of the 1C panels are close to 1B. We have created more distance between Figure 1B and 1C.

line 134: "others have observed" should have citations

Good point. Citation has been added.

line 170-174: the fact that reasonable filters cannot be fit to the data using reverse correlation because of limitations of the dataset does not mean that a filter-based description of the behavior could provide predictive power. Overall I don't think this section adds to the manuscript as filter and glm-based approaches have shown predictive power in other behavioral contexts.

Good point. We have reworded this section.

line 176: many GLMs incorporate time history, e.g.

Good point. We have reworded this section to specifically mention the GLM used in the study mentioned in the last review.

line 232: data limitations. I think this should be addressed in the Discussion as well. To what extent do failures of the model(s) or differences between conditions reflect data limitations versus more complexity?

Good point. Yes, we have now addressed this point in the Discussion.

line 264: KNN: please define

Done

line 286: weaving behavior: this is quite interesting and could be developed more.

Thanks! We have added a few more sentences.

line 323:329: these seem like Discussion points, not Results.

Although we understand the reviewer's point of view, we would like to keep those sentences in the results. We agree that these sentences are unnecessary for an expert. These sentences are necessary for readers not familiar with the anatomy of the *Drosophila* olfactory system.

line 447-451: "exact nature of the model" this specificity is what leads me to think that the current model will not be generally applicable, and might be better framed as an analysis rather than a model.

This comment was helpful in understanding the reviewer's earlier comment. We have added some sentences to the discussion to highlight this point.

line 608-610: short-term memory and dopamine. I think it would be helpful to break these two points up here. First the authors show evidence for short-term memory which is exciting and important. Second they speculate that it is due to dopamine which is a pure speculation based on the current data.

Great point! We have now stated that the dopamine comment is a speculation.

line 617: spatial sense of the stimulated arena. I think this is the first published evidence for this and the authors could draw more attention to it.

Thanks! We have done so.

Reviewer #2 (Remarks to the Author):

I appreciate the effort the authors have done towards addressing my concerns.

Thank you!

Reviewer #3 (Remarks to the Author):

The authors have carefully considered the alternative modeling options and appropriately substantiated their choices for the main text. I consider it reasonable to publish this upgraded version of the paper.

Thank you!